*GENETICS*, 2024, **227(1)**, iyae037

**Investigation**

# Spatio-temporal modeling of high-throughput multispectral aerial images improves agronomic trait genomic prediction in hybrid maize

Nicolas Morales (iD),[1] Mahlet T. Anche,[1] Nicholas S. Kaczmar,[1] Nicholas Lepak,[2] Pengzun Ni (iD),[1,3] Maria Cinta Romay (iD),[4] Nicholas Santantonio (iD),[1,5] Edward S. Buckler (iD),[1,2,4] Michael A. Gore (iD),[1] Lukas A. Mueller (iD),[1,6] Kelly R. Robbins (iD)[1,*]

[1]Plant Breeding and Genetics Section, School of Integrative Plant Science, Cornell University, Ithaca, NY 14853, USA
[2]United States Department of Agriculture-Agricultural Research Service, Robert W. Holley Center for Agriculture and Health, Ithaca, NY 14853, USA
[3]College of Bioscience and Biotechnology, Shenyang Agricultural University, Shenhe District, Shenyang, Liaoning Province, PR China
[4]Institute for Genomic Diversity, Cornell University, Ithaca, NY 14853, USA
[5]School of Plant and Environmental Sciences, Virginia Tech, Blacksburg, VA 24061, USA
[6]Boyce Thompson Institute, Ithaca, NY 14853, USA

*Corresponding author: Plant Breeding and Genetics Section, School of Integrative Plant Science, 102b Beebe Hall, Cornell University, 110 Arboretum Rd, Ithaca, NY 14850, USA. Email: krr73@cornell.edu

Design randomizations and spatial corrections have increased understanding of genotypic, spatial, and residual effects in field experiments, but precisely measuring spatial heterogeneity in the field remains a challenge. To this end, our study evaluated approaches to improve spatial modeling using high-throughput phenotypes (HTP) via unoccupied aerial vehicle (UAV) imagery. The normalized difference vegetation index was measured by a multispectral MicaSense camera and processed using ImageBreed. Contrasting to baseline agronomic trait spatial correction and a baseline multitrait model, a two-stage approach was proposed. Using longitudinal normalized difference vegetation index data, plot level permanent environment effects estimated spatial patterns in the field throughout the growing season. Normalized difference vegetation index permanent environment were separated from additive genetic effects using 2D spline, separable autoregressive models, or random regression models. The Permanent environment were leveraged within agronomic trait genomic best linear unbiased prediction either modeling an empirical covariance for random effects, or by modeling fixed effects as an average of permanent environment across time or split among three growth phases. Modeling approaches were tested using simulation data and Genomes-to-Fields hybrid maize (*Zea mays* L.) field experiments in 2015, 2017, 2019, and 2020 for grain yield, grain moisture, and ear height. The two-stage approach improved heritability, model fit, and genotypic effect estimation compared to baseline models. Electrical conductance and elevation from a 2019 soil survey significantly improved model fit, while 2D spline permanent environment were most strongly correlated with the soil parameters. Simulation of field effects demonstrated improved specificity for random regression models. In summary, the use of longitudinal normalized difference vegetation index measurements increased experimental accuracy and understanding of field spatio-temporal heterogeneity.

**Keywords:** unoccupied aerial vehicles; spatial correction; genomic prediction; vegetation indices; two-dimensional splines; random regression; autoregressive; permanent environment; high-throughput phenotypes; spatial heterogeneity; soil electrical conductance; elevation; soil curvature

## Introduction

Controlling for environmental heterogeneity in agricultural field experiments is critical to obtain accurate estimates of varietal performance and treatment effects (Van Es and Van Es 1993; Brownie *et al.* 1993; Smith *et al.* 2005; Xu 2016). In plant breeding where soil composition, elevation, slope, curvature, water content, nutrient availability, and management can vary within field experiments, the genotypic effects driving important agronomic traits can become confounded with a specific plot's permanent environment (PE) effects. In this context, a PE is a plot level nongenetic effect that is persistent across the growing season, giving rise to spatial patterns in the phenotypes of agronomic traits such as yield. Hereafter, PE will be used to describe nongenetic plot level

effects estimated using longitudinal data, and spatial effects will refer to nongenetic plot level effects estimated from agronomic data collected only at a single timepoint.

Randomization in experimental designs can help account for confounding genetic, spatial, and environmental variation to a large degree (Piepho *et al.* 2013; Hoefler *et al.* 2020), but in early stage trials where replication is limited, it is important to model spatial variation. Statistical approaches, such as the separable autoregressive process and the two-dimensional spline (2DSpl) model, have advanced to capture local dependence effects between experimental plots (Gilmour *et al.* 1997; Covarrubias-Pazaran 2016). Such spatial effects derive local dependencies from distance-based random covariance structures (e.g. plots that are close to each other are more interdependent than

those farther away), but these models often make simplifying assumptions of a consistent rate of decay in interdependency across the entire field. Nonetheless, modeling spatial effects using linear mixed models has improved experimental accuracy in plant breeding (Smith *et al.* 2005; Robbins *et al.* 2012; Rodríguez-Álvarez *et al.* 2018; Bernardeli *et al.* 2021; Copati *et al.* 2021).

Spatial heterogeneity can change over the growing season, due to weather and management conditions as well as plant development characteristics. The relationship of time on spatial effects can be explored using models accounting for covariance between timepoints and simple models nested within each timepoint. Repeated measurements in time allow estimation of PE effects from random regression (RR) models, providing a purely temporal representation of spatial heterogeneity. To effectively apply these statistical approaches, measurements should largely span the field; however, it can be difficult and expensive to objectively measure phenotypic traits repeatedly across a field experiment, as seen with quantitative disease resistance traits (Poland and Nelson 2011; Reynolds *et al.* 2019). Therefore, cost-effective remote sensing approaches are necessary for studying PE in the field across time.

Through the use of unoccupied aerial vehicles (UAVs) and other systems, aerial imaging can reliably and cost-effectively measure high-throughput phenotypes (HTPs) for all experimental plots in the field across the growing season (White *et al.* 2012; Andrade-Sanchez *et al.* 2014; Sagan *et al.* 2019; Sun *et al.* 2021). A widely studied class of aerial image HTPs are vegetation indices (VI) that include the normalized difference vegetation index (NDVI) (Gitelson *et al.* 2002; Hunt *et al.* 2013). VI provide physiologically relevant image features that can track variance such as photosynthetic activity, and have successfully measured chlorophyll content, canopy extent, biomass, and water-use efficiency among other plant attributes (Babar *et al.* 2006; Bannari *et al.* 2007; Delegido *et al.* 2011; Thorp *et al.* 2018). Promising model-derived HTPs from images exist, such as latent-space and convolutional neural network features; however, our study focused on estimation of a specific plot's NDVI PE from linear mixed models in order to quantify latent spatial heterogeneity and field environmental effects (Taghavi Namin *et al.* 2018; Gage *et al.* 2019; Wiesner-Hanks *et al.* 2019; Feldmann *et al.* 2021).

While phenotypic data are critical in plant breeding, genomic data are arguably of equal importance. Genomic best linear unbiased prediction (GBLUP) has been extensively applied to predict traits in animals and plants from genome-wide single nucleotide polymorphism (SNP) markers, including in maize and wheat (Meuwissen *et al.* 2001; Rutkoski *et al.* 2012; Daetwyler *et al.* 2013). GBLUP can predict traits from genome-wide SNP marker data by modeling the covariance of additive genetic effects as a genomic relationship matrix (GRM) (VanRaden 2008). Importantly for the models presented in this study, genomic relationships can improve the accuracy of partitioning genetic and nongenetic sources of variation, reducing potential confounding between additive genetic effects and plot level spatial heterogeneity.

VI can improve genomic prediction through multivariate approaches by leveraging genetic correlations between the VI and agronomic traits of interest, as demonstrated for grain yield in wheat and biomass in soybean (Rutkoski *et al.* 2016; Sakurai *et al.* 2021). While multitrait models leverage genetic correlations across traits to improve predictions, residual correlations exist between NDVI and grain yield in maize (Anche *et al.* 2020). Recent

studies have successfully proposed two-stage approaches for incorporating HTP, such as detecting spatial effects using the SpATS package in the first stage and then creating P-spline hierarchical growth models in the second stage (Pérez-Valencia *et al.* 2022). Furthermore, modeling approaches for integrating HTP, genomic information, and environmental information can be generalized into the genotype-to-phenotype model framework (van Eeuwijk *et al.* 2019); however, modeling NDVI PE for agronomic trait spatial corrections has not previously been widely explored and field tested.

Building on previous work, our study proposed a two-stage approach for improving agronomic trait spatial corrections. To summarize the two-stage approach, the first stage separated NDVI PE from additive genetic effects in the HTP, using either spatial corrections or RR models. The second stage summarized the NDVI PE within GBLUP for the agronomic traits using two distinct implementations, either modeling a plot-to-plot covariance of random effects (L) or fitting regressions on PE estimates from the first stage as fixed effects (FE). The proposed approach studied the following two questions utilizing simulated data and several years of hybrid maize field experiments. Firstly, are NDVI PE consistently able to detect spatial heterogeneity across the growing season affecting end-of-season agronomic traits? Secondly, can NDVI PE be used in the proposed two-stage models to improve spatial corrections for agronomic traits?

## Materials and methods
### Field experiments
As part of the Genomes-to-Fields (G2F) program, inbred and hybrid maize (*Zea mays* L.) field evaluations were planted at the Musgrave Research Farm (MRF) in Aurora, NY. Of importance to this study were the hybrid maize experiments planted in 2015 (Genomes to Fields 2020), 2017 (G2F Consortium 2019), 2019 (Genomes to Fields 2021), and 2020 (Genomes to Fields 2022), named 2015_NYH2, 2017_NYH2, 2019_NYH2, and 2020_NYH2, respectively. All experiments were randomized block designs and the tested hybrids varied across experiments (McFarland *et al.* 2020; AlKhalifah *et al.* 2018; Lima *et al.* 2023). The 2015_NYH2, 2017_NYH2, 2019_NYH2, and 2020_NYH2 experiments consisted of 375, 232, 612, and 401 unique hybrids, respectively, encompassing hybrids of diverse genetic backgrounds and expired-proprietary (Ex-PVP) biparental crosses.

The 2015_NYH2 biparental hybrids predominantly featured PHZ51, PHB47, LH82, and LH185 inbred testers and shared a common parent from three inbred biparental cross populations (PHN11 × PHW65, Mo44 × PHW65, and PHW65 × MoG). In 2019_NYH2 many biparental hybrids also shared a common parent from the PHN11 × PHW65 and Mo44 × PHW65 populations; however, PHT69 was the principal tester. The 2017_NYH2 and 2020_NYH2 experiments featured hybrids with shared parents from a MAGIC population (W10004) (Michel *et al.* 2022). In 2017_NYH2 the same 2015_NYH2 testers were predominantly featured along with 3IIH6 and PHRE1 inbreds; however, in 2020_NYH2, PHP02 was the principal inbred tester. Each experimental plot was seeded in two-row plantings. The 2015_NYH2, 2017_NYH2, 2019_NYH2, and 2020_NYH2 experiments were planted in 100 rows of 10, 10, 16, and 10 ranges, respectively, and plots were evenly distributed across two contiguous blocks in the field. In all field experiments, the following agronomic traits were measured: grain yield (GY) (bu/acre), grain moisture (GM) (%), and ear height (EH) (cm).

**Table 1.** Summary of UAV imaging dates for which HTP were successfully extracted in the 2015, 2017, 2019, and 2020 field experiments.

| Field experiment | Planting date | Field location | Growth stage: P0 | Growth stage: P1 | Growth stage: P2 |
|---|---|---|---|---|---|
| 2015_NYH2 | 2015 May 7 | MRF, Field P | July 21 | Aug 7, Aug 20 | Sept 10 |
| 2017_NYH2 | 2017 May 18 | MRF, Field Y | June 12 | Aug 2, Aug 17, Sept 1 | Sept 6, Sept 12, Sept 24 |
| 2019_NYH2 | 2019 May 23 | MRF, Field N | July 16, July 24, July 29 | Aug 5, Aug 15 | Sept 10 |
| 2020_NYH2 | 2020 May 22 | MRF, Field D | June 29, July 9, July 15, July 18 | July 22, July 28, Aug 1 | Aug 20, Aug 26, Sept 9, Sept 18, Oct 2 |

Growth stages of P0, P1, and P2 map to early, active, and late reproductive phases broadly defined as 0 to 1,225 GDD, 1,226 to 1,800 GDD, and 1,801 to 2,500 GDD, respectively.

Experimental variation across the years arose from the maize hybrids planted, the field sites utilized, the planting and harvest dates, the weather conditions experienced during the growing seasons, and the time points at which imaging events occurred. Supplementary Fig. 1 illustrates the time points at which HTPs were extracted, as well as the growing degree days (GDD) and cumulative precipitation (CP) on those days. GDD were calculated for each imaging event time point from the planting date of the experiment and were found using daily weather data for the ground station at MRF (GHCND:USC00300331) accessed via the NOAA NCEI NCDC database. The evidenced variation in GDD and CP across years highlighted the necessity to collect as many imaging events as possible over the growing season.

## Aerial image collection and processing

A MicaSense RedEdge 5-channel multispectral camera mounted onto an UAV captured images in the blue, green, red, near infrared, and red-edge spectra. The UAV flew at an altitude of 25 to 30 m and at a speed of 6 km/h. To complete a flight, the preprogrammed, serpentine flight plans required approximately 35 min to traverse the 3 km path. At least 80% overlap along both axes was ensured in the collected images.

Each flight produced approximately 5,000 images from the MicaSense camera, which were then processed into orthophotomosaics using Pix4dMapper photogrammetry software (Pix4D 2017). To produce reflectance calibrated raster images, the software used the MicaSense radiometric calibration panel images captured immediately prior to each UAV flight, as well as illumination metadata embedded in each capture by the MicaSense camera. Orthophotomosaic, or orthophoto, images were produced with approximately 1 cm per pixel resolution ground sample distance (GSD). The resulting reflectance orthophoto images were then uploaded into ImageBreed software, which enabled plot-polygon templates to be created and assigned to the field experimental design (Morales, Kaczmar, et al. 2020). Supplementary Fig. 2 illustrates a representative near-infrared (NIR) reflectance orthophoto image from 2019_NYH2 taken on August 15, 2019, with the plot-polygons overlaid. NDVI HTP were extracted through ImageBreed, and derived as the mean of NDVI pixel values within the plot boundaries (Gitelson et al. 2002; Hunt et al. 2013; Patrignani and Ochsner 2015; Bhandari et al. 2021). The image, field experiment, phenotypic, and genotypic data within ImageBreed are FAIR and can be queried through openly described APIs (Wilkinson et al. 2016; Selby et al. 2019).

Flights in 2015, 2017, and 2019 were scheduled approximately once per week, while 2020 targeted a frequency of twice per week. Technical problems and poor weather conditions, such as clouds, rain, and high winds, resulted in fewer imaging events being suitable for HTP extraction. To separate the HTP imaging event dates into biological growth stages, GDD ranges were defined to account for the early vegetative phase (P0) at 0 to 1,225 GDD, the active reproductive phase (P1) at 1,226 to 1,800 GDD,

and the late reproductive phase (P2) at 1,801 to 2,500 GDD. The three GDD ranges captured periods where NDVI was first steadily increasing, then plateauing, and finally steadily decreasing. Table 1 summarizes the growth stage distribution of imaging event dates for which HTP were successfully extracted.

## Soil information

In 2019 at the MRF, a ground conductivity meter (EM38-MK2, Geonics, Canada) surveyed Field N where the G2F hybrid maize field experiment was planted. The EM-38 probe used electrical inductance to characterize variation originating from a combination of factors including soil salinity, soil texture, water content, water retention, soil type, and soil nutrients (Heil and Schmidhalter 2017). The goals for the soil information in this study were to (1) better understand driving factors for the estimated NDVI PE from the aerial imagery, and (2) determine whether a soil survey was a practical alternative to aerial imagery for improving agronomic spatial corrections in the second stage. Due to planting rotations and other logistical concerns, the G2F experiments conducted in 2015, 2017, 2019, and 2020 were all in distinct field locations as shown in Table 1; therefore, the soil information in this study could only be applied to the 2019 experiment.

The soil survey was conducted prior to the hybrids being planted by passing the probe over the field in a dual-serpentine pattern, with nine passes in the east–west orientation and 24 passes in the north–south orientation. The georeferenced elevation (Alt) and apparent electrical conductance (EC) data were then interpolated over the entire field using ordinary Kriging in R (Pebesma 2004). The interpolated raster was produced using a spherical variogram model at a resolution of $10^{-6}$ WGS84 units covering 120 by 200 cells. Supplementary Fig. 3 illustrates (A) a map of the collected EM38 soil survey, (B) the region to interpolate into, and in (C) and (D) the interpolated soil EC and Alt across the field, respectively. Finally, a mean value for the soil EC and Alt was extracted using ImageBreed for each plot in the 2019_NYH2 field experiment.

In order to approximate soil curvature as elevation gradients, first and second 2D numerical derivatives were computed on the plot level soil EC and altitude measurements, denoted as dEC, d2EC, dAlt, and d2Alt, respectively. 2D numerical derivatives were computed by averaging the differences between a given plot and the three immediately adjacent rings encircling it. Supplementary Fig. 4 illustrated heatmaps of the extracted plot level soil EC and Alt, along with the first and second derivatives.

## Genotyping data

G2F maize inbred lines were scored with a genotyping-by-sequencing (GBS) approach that yielded 945,574 SNP markers across the genome for 1,577 samples representing a total of 1,325 unique maize inbred lines (G2F Consortium 2019) (Elshire et al. 2011; McFarland et al. 2020). The resulting genome-wide

variant call format (VCF) data were queried for the hybrids in the 2015, 2017, 2019, and 2020 field experiments (Danecek *et al.* 2011; Morales, Bauchet, *et al.* 2020; Morales *et al.* 2022). Due to minor typographical errors (e.g. Mo17 vs MO17), data cleaning was required prior to mapping the sample identifiers in the VCF to the field experiment genotype identifiers and to the pedigree information for the maize hybrids. Genotypes were filtered for SNPs with minor allele frequency <5% or with >40% missing data, and for samples containing >20% missing data. GRMs were computed using the A.mat function in rrBLUP, with missing data imputed as the mean genotype (VanRaden 2008; Endelman 2011). In this study, only additive genetic relationships were modeled (Griffing 1956).

Given that many of the evaluated maize hybrids in the G2F program originated as bi-parental crosses of the genotyped inbred lines and that the inbred pollen and seeds parents are unrelated to each other, hybrid maize genotypes were computed from parental GRMs following:

$$r_{ij} = 0.5 * (r_{m_i,m_j} + r_{p_i,p_j})$$

where $r_{ij}$ is the genomic relationship between hybrids, $r_{m_i,m_j}$ is the genomic relationship between the seed parents of the hybrids, and $r_{p_i,p_j}$ is the genomic relationship between the pollen parents of the hybrids. Therefore, the hybrid genotype is computed by averaging the parents' genomic relationships for general combining ability (GCA), where the parents' genomic GCA matrix was calculated as:

$$\mathbf{G_{GCA}} = 0.5 * \left( \frac{\mathbf{W}t(\mathbf{W})}{2 \sum p_i(1 - p_i)} \right)$$

where $\mathbf{W}$ is the mean centered allelic dosage matrix (2—homozygous reference allele, 1—heterozygous, 0—homozygous nonreference allele) for the inbred parents and $p_i$ is the frequency of the ith SNP marker. If a hybrid's parental inbred lines were not genotyped, then the hybrid was included in the GRM with a diagonal value of one and off-diagonal values of zero.

## HTP spatial heterogeneity

The first question of this study was whether a specific plot's non-genetic NDVI PE, which were estimated by spatial and temporal RR effects, consistently detected across the growing season the spatial heterogeneity affecting end-of-season agronomic traits. Therefore, the first stage of the proposed two-stage approach focused on measuring spatial heterogeneity in NDVI across the growing season. All analyses were nested within year given the limited overlap in entries between years.

Variance in the collected HTP observations, $V_P$, was modeled as arising from genetic variance among the maize hybrids, $V_G$, and from environmental sources of variance, $V_E$ (Falconer and Mackay 1996). In mixed model matrix notation this was formulated as

$$y = \mathbf{X}\beta + \mathbf{Z_a}u_a + \mathbf{Z_p}u_p + e \tag{1}$$

where $y$ is a vector of phenotypic observations and $\mathbf{X}$ is an incidence matrix mapping phenotypic values to the fixed effects, $\beta$, for replicate (nested within timepoint for analyses including multiple timepoints). The random effects $u_a$ and $u_p$ represented the additive genetic and the specific plot's PE, respectively. The incidence matrices $\mathbf{Z_a}$ and $\mathbf{Z_p}$ linked the random effects $u_a$ and $u_p$ to

the observed phenotypes. The random residual error is represented by $e$.

The following paragraphs detail how Equation (1) modeled the variation of NDVI PE across time by utilizing either a single HTP time point or many HTP time points simultaneously.

The spatial model fitted either a single HTP time point or multiple HTP time points, a distinction referred to as the single-trait or single-trait-repeated cases. In the single-trait case, spatial models were run independently for each of the collected HTP time points. In contrast, the single-trait-repeated case fitted several collected HTP time points in a single model, such that the vector y represented HTP observations taken at different time points across the growing season:

$$y = \begin{bmatrix} y_{t1} \\ y_{t2} \\ \vdots \end{bmatrix}$$

In the single-trait spatial case, the variance of the random additive genetic effect was defined as:

$$\text{var}(u_a) = \sigma_{u_a}^2 \mathbf{G} \tag{2}$$

where the matrix $\mathbf{G}$ represented the GRM between evaluated hybrids (VanRaden 2008). The residual variance was modeled as:

$$\text{var}(e) = \sigma_e^2 \mathbf{I}$$

where $\mathbf{I}$ is the identity matrix.

The single-trait-repeated case defined this as

$$\text{var}(u_a) = \mathbf{\Sigma_{u_a}} \otimes \mathbf{G} = \begin{bmatrix} \sigma_{a_1}^2 & \sigma_{a_1}\sigma_{a_2} & \cdots \\ \sigma_{a_2}\sigma_{a_1} & \sigma_{a_2}^2 & \cdots \\ \vdots & \vdots & \ddots \end{bmatrix}_{[\mathbf{t,t}]} \otimes \mathbf{G} \tag{3}$$

where the unstructured matrix $\mathbf{\Sigma_{u_a}}$ is of order $[t, t]$ denoting the number of time points involved and $\otimes$ is the Kronecker product.

$$\text{var}(e) = \mathbf{\Sigma}_e \otimes \mathbf{I} = \begin{bmatrix} \sigma_{e_1}^2 & \sigma_{e_1}\sigma_{e_2} & \cdots \\ \sigma_{e_2}\sigma_{e_1} & \sigma_{e_2}^2 & \cdots \\ \vdots & \vdots & \ddots \end{bmatrix}_{[t,t]} \otimes \mathbf{I}$$

where the unstructured matrix $\mathbf{\Sigma}_e$ is of order $[t, t]$ denoting the number of time points involved, $\mathbf{I}$ is an identity matrix with dimensions equal to the number of plots, and $\otimes$ is the Kronecker product. The advantage of the single-trait-repeated approach was to explicitly model genetic and residual covariances between time points. When model convergence became problematic due to high numbers of time-points in Equation (3), a minimum of three time-points were selected based on NDVI heritability and correlation to yield.

Two methods for modeling environmental spatial variation were investigated, namely 2DSpl and AR1 models. The 2DSpl method used penalized splines fitted using sommer in R (R 3.6.3, Sommer 4.1.3) (Covarrubias-Pazaran 2016). The AR1 method explicitly defined a separable autoregressive covariance structure fitted using ASReml R (R 3.6.3, ASReml R 4.1.0.126) (Gilmour et al. 2015). These two methods use row and column information to model spatial heterogeneity and are further explained in Supplementary File 2. As an alternative to spatial mixed models, random regression (RR) models were explored for separating NDVI PE from additive genetic effects and residual variation

specific to a given timepoint (Kirkpatrick *et al.* 1990; Van der Werf *et al.* 1998; Schaeffer 2004; Arnold *et al.* 2019).

RR PE offered a purely longitudinal representation of the plot level spatial heterogeneity in the field. The RR model notably allows estimation of additive genetic effects and PE effects whose covariance follows the GRM and a plot-to-plot correlation matrix **E**, respectively. The RR model can be expressed in mixed model matrix notation as in Equation (1), however, the incidence matrices $\mathbf{Z_a}$ and $\mathbf{Z_p}$ contain Legendre polynomial functions of the time, in GDD, at which the measurement was recorded, and $u_a$ and $u_p$ denote the hybrid specific random additive genetic and PE regression coefficients, respectively. The fixed effect was for the design replicate nested with the imaging event date, year, and location, while the random residual $e$ whad heterogeneous variance. The overall variance was written as:

$$\mathrm{var}(y) = \mathbf{Z_a}(\Sigma_{\mathbf{u_a}} \otimes \mathbf{G})\mathbf{Z'_a} + \mathbf{Z_p}(\Sigma_{\mathbf{u_p}} \otimes \mathbf{E})\mathbf{Z'_p} + \Sigma_\mathbf{e} \otimes \mathbf{I}$$

In this equation, **G** represents the GRM and **E** represents a plot-to-plot covariance matrix capturing environmental effects, ideally computed from envirotyping information. Envirotyping aimed to uniquely define the complete environment of an organism by including soil, climate, and developmental parameters; however, this study focused on applying NDVI PE (Xu 2016).

The additive genetic variance for any hybrid at a specific time point ($\sigma^2_{a,t_i}$) and the additive genetic covariance for any hybrid between two time points ($\sigma_{a,t_it_j}$) can be calculated using:

$$\sigma^2_{a,t_i} = z^T_{t_i}\Sigma_{u_a}z_{t_i}$$

and

$$\sigma_{a,t_it_j} = z^T_{t_i}\Sigma_{u_a}z_{t_j}$$

where $z_{t_i}$ and $z_{t_j}$ are vectors of the continuous random regression function evaluated at time points $t_i$ and $t_j$, respectively. Similar expressions for the PE variance ($\sigma^2_{p,t_i}$) and covariance ($\sigma_{p,t_it_j}$) can be written as

$$\sigma^2_{p,t_i} = z^T_{t_i}\Sigma_{\mathbf{u_p}}z_{t_i}$$

and

$$\sigma_{p,t_it_j} = z^T_{t_i}\Sigma_{\mathbf{u_p}}z_{t_j}$$

The residual variance structure :

$$\mathrm{var}(e) = \Sigma_e \otimes \mathbf{I} = \begin{bmatrix} \sigma^2_{e_1} & 0 & \cdots \\ 0 & \sigma^2_{e_2} & \cdots \\ \vdots & \vdots & \ddots \end{bmatrix}_{[t,t]} \otimes \mathbf{I}$$

where the heterogeneous diagonal matrix $\Sigma_e$ is of order $[t, t]$ denoting the number of time points involved, **I** is an identity matrix with dimensions equal to the number of plots, and $\otimes$ is the Kronecker product. In this study, solutions to the RR model were estimated using the BLUPF90 family of programs (Misztal *et al.* 2002).

The RR model had computational benefits over the spatial mixed models described previously. Firstly, the number of variance components to estimate was equal to $\frac{1}{2}n(n-1) + n$, where $n$ is the order of the respective random regression functions,

regardless of the number of time points $t$ represented in the observations y. Secondly, continuous curves for the random additive genetic and PE effects could be evaluated for any time point because the random regression coefficients fit a covariance function. Thirdly, there was flexibility in the type of regression function that can be fitted, for instance, splines vs Legendre polynomials (Szeg 1975). In this study, third-order Legendre polynomials were considered after looking at the log-likelihood of first- and second-order polynomials as well as the correlation of HTP PE effects and agronomic trait spatial heterogeneity. Furthermore, third-order Legendre polynomials have been found to fit multi-spectral VI well (Anche *et al.* 2023). Fourthly, flexible specification of **E** allowed envirotyping information to be accounted for in the model.

This study explored six structures for the RR PE covariance matrix **E**. The first approach, named RRID, defined **E** = **I**, where **I** is the identity matrix. The second approach, named RREuc, computed **E** using the inverse Euclidean distances between experimental plots and standardized between 0 and 1. This is written as

$$E_{ij} = \frac{1}{\sqrt{(r_i - r_j)^2(c_i - c_j)^2}}$$

where $E_{ij}$ is an element of **E** denoting the relationship between plot $i$ and $j$, and the variables $r_i$, $c_i$, $r_j$, and $c_j$ are the row and column ordinal positions of the plot, respectively. The remaining approaches named RRSoilEC, RRSoilAlt, RR2DSpl, and RRAR1 computed **E** following:

$$\mathbf{E} = \frac{\mathbf{QQ'}}{m}$$

where **Q** is a centered and standardized matrix of the considered features and $m$ is the number of distinct features. The plot level features in these cases were values of soil EC, dEC, and d2EC, values of soil Alt, dAlt, and d2Alt, values of 2DSplM NDVI random spatial effects, and values of AR1M NDVI random spatial effects, respectively.

The presented models allowed estimation of additive genetic and PE random effects; however, the true genetic and environmental effects were unknown to us. Therefore, to evaluate the robustness of the tested models and their ability to detect field environment features, six different simulation scenarios named Linear, 1D-N, 2D-N, AR1xAR1, and RD were conducted. The simulation methods are described in Supplementary File 1 and were designed to represent purely environmental effects in the field, such as due to soil heterogeneity, altitude, and soil elevation gradients. Each were tested by varying the correlation across time to be 0.75, 0.90, and 1.00 and by setting the simulated variance to be 10%, 20%, and 30% of the total phenotypic variance. Evaluating the accuracy of the simulation process followed:

1. For a target model, NDVI PE were separated from additive genetic effects present in the real NDVI HTP for a given field experiment.
2. The computed NDVI PE were subtracted from the NDVI HTP in order to minimize latent spatio-temporal effects in the NDVI.
3. The target simulation values, meaning one of the six simulation processes, were scaled between 0 and 1, then subsequently scaled to account for either 10%, 20%, or 30% of

the observed NDVI phenotypic variation, and finally were added onto the minimized NDVI HTP.

4. The target model computed PE for the simulation-adjusted NDVI HTP.
5. Finally, the recovered PE were correlated against the true target simulation values, returning prediction accuracy.

## Agronomic spatial correction

The second question in this study was whether the longitudinal NDVI or NDVI PE data could be used to improve spatial corrections for genomic prediction of agronomic traits. Firstly, the following baseline models were defined under the GBLUP framework. The baseline GBLUP model was written as:

$$y = \mathbf{X}\beta + \mathbf{Z_a}u_a + e \tag{4}$$

where y is the agronomic trait of interest, $\beta$ is a vector of fixed effects for design replication nested by year and location, $u_a$ is a vector of the random additive genetic effects of the hybrids, $\mathbf{X}$ and $\mathbf{Z_a}$ were incidence matrices linking model terms to y, and e was a vector of the residual errors. The variance of the random additive genetic effect was defined as:

$$\text{var}(u_a) = \sigma_{u_a}^2 \mathbf{G}$$

where $\mathbf{G}$ is the GRM and $\sigma_{u_a}^2$ is the additive genetic variance. The error variance is defined as:

$$\text{var}(e) = \sigma_e^2 \mathbf{I}$$

where $\sigma_e^2$ is the residual variance. Control GBLUP models named G, G + 2DSpl, and G + AR1, are as follows:

$$y = \mathbf{X_R}\beta_R + \mathbf{Z_a}u_a + e \tag{5a}$$

$$y = \mathbf{X_R}\beta_R + \mathbf{Z_a}u_a + \mathbf{Z_{2DSpl}}u_{2DSpl} + e \tag{5b}$$

$$y = \mathbf{X_R}\beta_R + \mathbf{Z_a}u_a + \mathbf{Z_{AR1}}u_{AR1} + e \tag{5c}$$

These models are used as baseline models for agronomic genomic prediction, and defined the vector $\beta_R$ for fixed effects of replication nested by year and location corresponding to incidence matrix $\mathbf{X_R}$, while e represented the residual errors. Equation (5a) is the simplest GBLUP case, modeling only random genetic effects $u_a$ with corresponding incidence matrix $\mathbf{Z_a}$. Equations (5b) and (5c) added 2DSpl and AR1 random effects, $u_{2DSpl}$ and $u_{AR1}$, respectively, with corresponding incidence matrices $\mathbf{Z_{2DSpl}}$ and $\mathbf{Z_{AR1}}$. Incorporating spatial corrections accounts for the row and column positions of the plots in the field on which the agronomic trait was measured. Equation (5b) followed the 2DSpl definition from Equation (S1), while Equation (5c) followed the AR1 definition from Equation (S2). Importantly, these three baseline models did not leverage information from the first stage or from the aerial image HTP measurements.

A final baseline defined the multitrait model (M) in Equation (6),

$$y = \mathbf{X_{RT}}\beta_{RT} + \mathbf{Z_a}u_a + e \tag{6}$$

where y is a vector of both the HTP NDVI time points and the agronomic trait (e.g. GY or EH or GM), $\beta_{RT}$ is a vector for fixed effects of replicate nested with trait, year, and location, $u_a$ is a vector of the

random additive genetic effects for all traits, and e is the random residual variance. The random additive genetic ($u_a$) and residual (e) covariances are unstructured across the traits, as in Equation (3). Given the difficulty of fitting large numbers of traits in M, only the two HTP NDVI timepoints with the highest correlation to GY were included in y.

Secondly, to improve on the baseline GBLUP models first-stage, NDVI PE were integrated into the second stage following two implementations. The first implementation modeled the NDVI PE as a plot-to-plot correlation structure for random effects (L), following:

$$y = \mathbf{X_R}\beta_R + \mathbf{Z_a}u_a + \mathbf{Z_p}u_p + e \tag{7}$$

The variance of the random plot effects $u_p$ is:

$$\text{var}(u_p) = \sigma_p^2 \mathbf{L}$$

where:

$$\mathbf{L} = \frac{\mathbf{NN}'}{t}$$

The matrix $\mathbf{N}$ is the centered and standardized NDVI PE and t is the number of time points considered. The vector $\beta_R$ is for the fixed effects of replicate nested with imaging event date, year, and location, with corresponding incidence matrix $\mathbf{X_R}$. This model is similar to Equation (5c) in which the AR1 process explicitly defined a plot-to-plot covariance structure; however, rather than define distance-based assumptions, Equation (7) utilized observed spatial heterogeneity. The second column of Table 2 summarizes the tested L two-stage models in relation to the first-stage.

Alternatively, fixed FE were defined and followed four variations:

$$y = \mathbf{X_R}\beta_R + \dot{\beta}_{H_{avg}} H_{avg} + \mathbf{Z_a}u_a + e \tag{8a}$$

$$y = \mathbf{X_R}\beta_R + \beta_{\dot{H}_1} H_1 + \beta_{\dot{H}_2} H_2 + \beta_{\dot{H}_3} H_3 + \mathbf{Z_a}u_a + e \tag{8b}$$

$$y = \mathbf{X_R}\beta_R + \beta_{\dot{F}_{avg}} F_{avg} + \mathbf{Z_a}u_a + e \tag{8c}$$

$$y = \mathbf{X_R}\beta_R + \beta_{\dot{F}_1} F_1 + \beta_{\dot{F}_2} F_2 + \beta_{\dot{F}_3} F_3 + \mathbf{Z_a}u_a + e \tag{8d}$$

In Equation (8a), the vector $H_{avg}$ performed a continuous regression on the average first-stage NDVI PE across all time points, resolving the $\beta_{H_{avg}}$ coefficient. Alternatively, Equation (8b) performed a continuous regression on the vectors $H_1$, $H_2$, and $H_3$ by splitting time points into the previously defined P0, P1, and P2 growth phases, respectively, and then averaging the first-stage NDVI PE within each. This model resolves coefficients $\beta_{\dot{H}_1}$, $\beta_{\dot{H}_2}$, and $\beta_{\dot{H}_3}$ for the P0, P1, and P2 growth phases, respectively. Equations (8c) and (8d) used binned fixed effects derived by reassigning the first-stage NDVI PE to quartile factors (1 = 0–25%, 2 = 26–50%, 3 = 51–75%, 4 = 76–100%), representing poor, marginal, good, and high performing levels. The $F_{avg}$ vector in Equation (8c) was computed by averaging over all time points, resolving the $\beta_{\dot{F}_{avg}}$ coefficient. The $F_1$, $F_2$, and $F_3$ vectors in Equation (8d) were computed by splitting the time points into the P0, P1, and P2 growth phases, respectively, resolving $\beta_{\dot{F}_1}$, $\beta_{\dot{F}_2}$, and $\beta_{\dot{F}_3}$ coefficients. Columns 3 to 6 of Table 2 summarize the tested FE two-stage model names in relation to the first-stage.

**Table 2.** Listed are all nonsoil two-stage models tested in this study.

| Stage-1 model[a] | Stage-2 L model | Stage-2 $H_{avg}$ model | Stage-2 H3 model | Stage-2 $F_{avg}$ model | Stage-2 F3 model |
|---|---|---|---|---|---|
| 2DSplU | G + L_2DSplU | G + Havg_2DSplU | G + H3_2DSplU | G + Favg_2DSplU | G + F3_2DSplU |
| 2DSplM | G + L_2DSplM | G + Havg_2DSplM | G + H3_2DSplM | G + Favg_2DSplM | G + F3_2DSplM |
| AR1U | G + L_AR1U | G + Havg_AR1U | G + H3_AR1U | G + Favg_AR1U | G + F3_AR1U |
| AR1M | G + L_AR1M | G + Havg_AR1M | G + H3_AR1M | G + Favg_AR1M | G + F3_AR1M |
| RRID | G + L_RRID | G + Havg_RRID | G + H3_RRID | G + Favg_RRID | G + F3_RRID |
| RREuc | G + L_RREuc | G + Havg_RREuc | G + H3_RREuc | G + Favg_RREuc | G + F3_RREuc |
| RR2DSpl | G + L_RR2DSpl | G + Havg_RR2DSpl | G + H3_RR2DSpl | G + Favg_RR2DSpl | G + F3_RR2DSpl |
| RRAR1 | G + L_RRAR1 | G + Havg_RRAR1 | G + H3_RRAR1 | G + Favg_RRAR1 | G + F3_RRAR1 |

The first column lists the first-stage models used to separate additive genetic effects from NDVI PE effects. Subsequent columns list models where the second-stage was implemented as a plot-to-plot covariance (L), as a continuous regression (Havg), as three distinct continuous fixed effects (H3), as three distinct binned fixed effects (F3), and as the average of the three binned effects ($F_{avg}$).
[a]2DSplU, two-dimensional spline spatial model fit to single timepoints; 2DSplM, two-dimensional spline spatial model fit to multiple timepoints; AR1U, first-order autoregressive spatial model fit to single timepoints; AR1M, first-order autoregressive spatial model fit to multiple timepoints; RRID, random regression using an identity matrix to model plot-to-plot correlations for PE; RREuc, random regression using Euclidian distance to model plot-to-plot correlations for PE; RR2DSpl, random regression using 2DSpl spatial effect estimates to model plot-to-plot correlations for PE; RRAR1, random regression using first-order autoregressive spatial effect estimates to model plot-to-plot correlations for PE.

Soil altitude and EC were measured in 2019, enabling Equations (7), (8a) and (8c) to account for soil information rather than first-stage NDVI PE. It was not possible to test Equation (8b) or (8d) with soil information because only a single soil measurement was collected. Representing Equation (7), a model named G + L_RRSoilEC constructed $L$ using soil EC, dEC, and d2EC features, while a model named G + L_RRSoilAlt constructed $L$ using soil Alt, dAlt, and d2Alt features. Using soil measurements in Equation (8a), models named G + Havg_Soil_Alt, G + Havg_Soil_dAlt, G + Havg_Soil_d2Alt, G + Havg_Soil_EC, G + Havg_Soil_dEC, and G + Havg_Soil_d2EC represented $H_{avg}$ derived from the plot level soil Alt, dAlt, d2Alt, EC, dEC, and d2EC, respectively. Whereas for Equation (8c), models named G + Favg_Soil_Alt, G + Favg_Soil_dAlt, G + Favg_Soil_d2Alt, G + Favg_Soil_EC, G + Favg_Soil_dEC, and G + Favg_Soil_d2EC represented $F_{avg}$ derived from the plot level soil Alt, dAlt, d2Alt, EC, dEC, and d2EC, respectively.

To measure the impact of spatial corrections on plot level data points, the plot level heritability is calculated as:

$$h^2 = \frac{\sigma^2_{u_a}}{\sigma^2_{u_a} + \sigma^2_e}$$

where $\sigma^2_{u_a}$ is the additive genetic variance component and $\sigma^2_e$ is the residual error variance component.

Model fit and genotypic effect estimation across replicates (GEER) were used to measure model accuracy. Model fit was defined as the correlation between fitted model predictions $\hat{y}$ and the true phenotypic values y, written as cor($\hat{y}$, y). Finally, GEER, written as cor($g_{rep1}$, $g_{rep2}$), involved first partitioning the agronomic trait datasets by replicate, then running the target model on each replicate, and finally correlating the random genetic effect estimates, $u_a$, across the replicates. Given mixed model analyses were performed with varying covariance structures, entry mean heritability was not used as a measure of accuracy.

## Results and discussion
### Local environmental effects
Before turning attention to the NDVI HTP, spatial corrections for the agronomic traits of GY, GM, and EH were computed. As defined in Equation (5b), Fig. 1 illustrates heatmaps of the 2DSpl spatial effects over the rows and columns of the experimental plots in the 2017_NYH2, 2019_NYH2, and 2020_NYH2 field experiments. Each year the trial was planted in a distinct field location. The

heatmaps resolved major, poorly performing regions for GY and EH centered around the row-column positions of (35,2), (19,8), and (25,6) in 2017_NYH2, 2019_NYH2, and 2020_NYH2, respectively. For all three experiments, an inverse spatial pattern was visible for GY and GM, while a similar spatial pattern was visible for GY and EH. The same pattern between traits was evidenced in 2015_NYH2, as illustrated in Supplementary Fig. 7. In 2015_NYH2 a poor performing region for GY and EH was centered near the row-column position of (88,9). In all four years, the proportion of phenotypic variation explained by the 2DSpl spatial effect ranged from ±25 bu/acre of GY, ±2% of GM, and ±15 cm of EH. These results illustrated the importance of spatial heterogeneity on the agronomic traits.

Similar patterns of spatial random effects were found using the 2DSpl and AR1 models defined in Equations (5b) and (5c), respectively. Table 3 lists the correlations between the 2DSpl and AR1 spatial effects for GY, GM, and EH individually in the 2017_NYH2, 2019_NYH2, and 2020_NYH2 experiments. The strongest average correlation was 0.88 for EH, followed by 0.87 for GY, and 0.49 for GM. The 2015_NYH2 experiment was not included in Table 3 because the AR1 model did not converge.

To understand correspondence between PE affecting NDVI and the end-of-season agronomic traits, first-stage NDVI PE were compared to GY, GM, and EH spatial effects. Figure 2 shows correlations between the 2DSpl GY spatial effects and the 2DSplU NDVI PE across 12 time points in the 2020_NYH2 field experiment. Illustrated in Fig. 2, correlations at 38 and 48 days after planting (DAP) were 0.6 and 0.7, respectively, and correlations fluctuated between 0.5 and 0.7 throughout the season. Corresponding heatmaps in Fig. 2 illustrate spatial distributions over the rows and columns of the experimental plots, and consistently reveal a large region near the center of the field negatively impacting both GY and NDVI. For reference, phenotypic correlations between NDVI and GY in 2020_NYH2 ranged from a low of 0.04 at 133 DAP to a high of 0.39 at 54 DAP, which were weaker than the correlations observed in Fig. 2. Similarly, the RR model PE effects correlated with the GY spatial effects more strongly than the NDVI and GY themselves. Supplementary Fig. 8 illustrates RRID PE effects correlated against GY and the GY 2DSpl and AR1 spatial effects. The RRID PE tended to be strongly correlated through time and identified similar spatial patterns as the spatial effect models.

In 2019_NYH2, 2015_NYH2, and 2017_NYH2 similar correlations between 2DSplU and GY spatial effects were observed, as illustrated in Fig. 3, Supplementary Figs. 9, and 10, respectively. For reference, phenotypic correlations between NDVI and GY in 2015_NYH2, 2017_NYH2 and 2019_NYH2 ranged from a low of

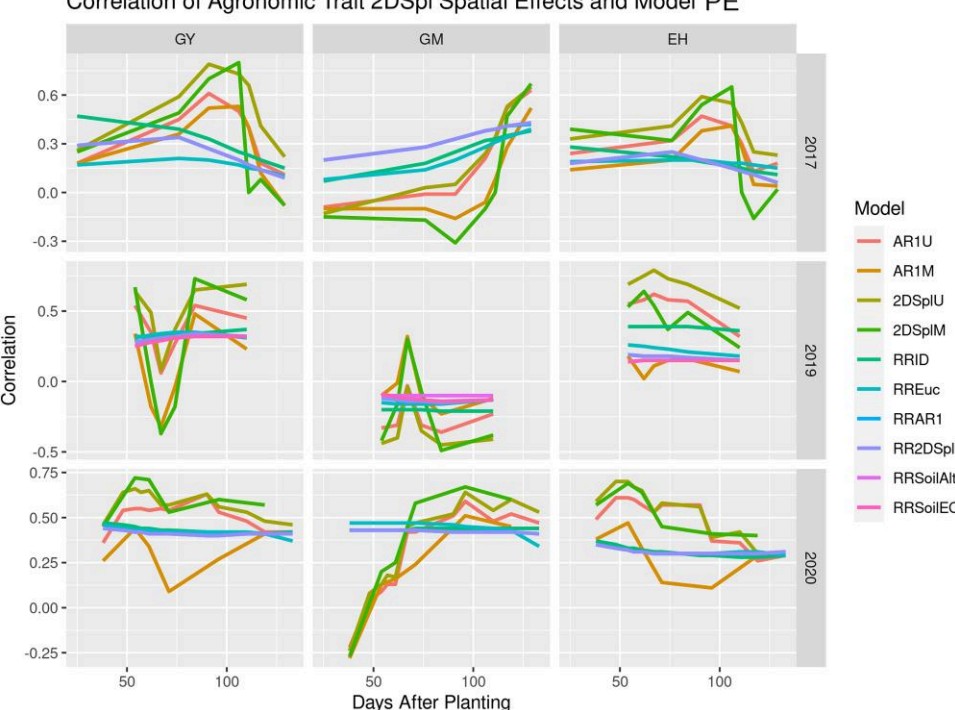

**Fig. 1.** 2DSpl spatial random effects detected in the agronomic traits of GY, GM, and EH in the 2017, 2019, and 2020 field experiments. The 2015 experiment is shown in Supplementary Fig. 7.

**Table 3.** Correlations between 2DSpl and first-order autoregressive (AR1) spatial effects for GY, GM, and EH individually in the 2017, 2019, and 2020 field experiments.

| Experiment | Grain yield (GY) | Grain moisture (GM) | Ear height (EH) |
|---|---|---|---|
| 2017_NYH2 | 0.90 | 0.49 | 0.74 |
| 2019_NYH2 | 0.91 | 0.59 | 0.93 |
| 2020_NYH2 | 0.79 | 0.40 | 0.97 |

0.13 at 126 DAP, 0.17 at 25 DAP, and 0.33 at 110 DAP, respectively, to a high of 0.42 at 92 DAP, 0.49 at 91 DAP, and 0.67 at 84 DAP, respectively. Therefore, in all experiments the NDVI PE were more correlated to the GY 2DSpl effects across the growing season than the NDVI were correlated to GY. Furthermore, in all years, the spatial patterns affecting GY and EH were detectable by NDVI PE to a large degree (>0.5 correlation), evidenced early on in the growing season at 76 DAP or less.

As a potential alternative to aerial imaging and to better understand the observed spatial effects, soil EC, Alt, and the first and second 2D numerical derivatives (dEC, d2EC, dAlt, d2Alt) were compared with the NDVI PE in the 2019_NYH2 field experiment. Figure 3 includes correlations and heatmaps of the soil measurements with the spatial effects and demonstrated: (1) the 2DSpl spatial effects of GY in 2019_NYH2 correlated up to 0.7 with 2DSplU NDVI PE, (2) soil EC correlated up to 0.5 with 2DSplU NDVI PE, and (3) soil d2Alt correlated up to 0.3 with 2DSplU NDVI PE. In Fig. 3, the soil elevation gradients, represented by heatmaps of Alt, EC, and their derivatives, highlighted the contours of the observed 2DSpl spatial effects. Supplementary Fig. 11 presents an analog to Fig. 3 showing AR1U NDVI PE, GY, and GY AR1 spatial effects in 2019_NYH2, and showed (1) overall weaker correlations between the GY AR1 spatial effects and the AR1U NDVI PE, with a high of 0.5, (2) similar correlations to soil

EC with a high of 0.5, and (3) overall weaker correlations to soil d2Alt with a high of 0.2. Therefore, the AR1U NDVI PE were less correlated with the soil parameters than the 2DSplU NDVI PE, and the AR1U NDVI PE were less correlated with GY AR1 effects than the 2DSplU NDVI PE were correlated with the GY 2DSpl effects.

Table 4 summarizes correlations between the soil information in 2019_NYH2 and the model NDVI PE, illustrating how the average first-stage NDVI PE across all time points correlated to soil EC, dEC, d2EC, Alt, dAlt, and d2Alt. Table 4 indicates 2DSplU had the strongest correlation to EC of 0.46 and also correlated relatively strongly with d2Alt. The d2Alt tended to correlate more strongly than Alt or dAlt with the NDVI PE, indicating the importance of elevation gradients in the field. The correlations between 2DSpl NDVI PE and soil parameters indicated that soil information was capturing similar spatial information as the NDVI aerial imaging.

Summarizing results between the agronomic trait 2DSpl effects and the tested first-stage model PE, Fig. 4 presents correlations for all agronomic traits and years. Figure 4 indicates the 2DSplU, 2DSplM, and AR1U models produced NDVI PE most correlated to the 2DSpl effects of GY and EH in all years and in nearly all time points, while the RR2DSpl, RRAR1, and RRSoilEC models were most correlated to the 2DSpl effects of GM in 2015, 2017, and 2019. The 2DSplU and AR1U NDVI PE correlated with GY and EH 2DSpl spatial effects greater than 0.5 in all years at 80 to 90 DAP. Significant similarities were seen between models run on traits within a given year, for instance both GY and EH in 2017 showed a large peak at 110 DAP and in 2020 both showed a continuous gradual decline. There was an inverted behavior between the GY and GM spatial effects in all years, describable as: in 2015 a high for GY and a low for GM at 105 DAP, in 2017 a high for GY and a low for GM at 110 DAP, in 2019 a high for GY and a low for GM

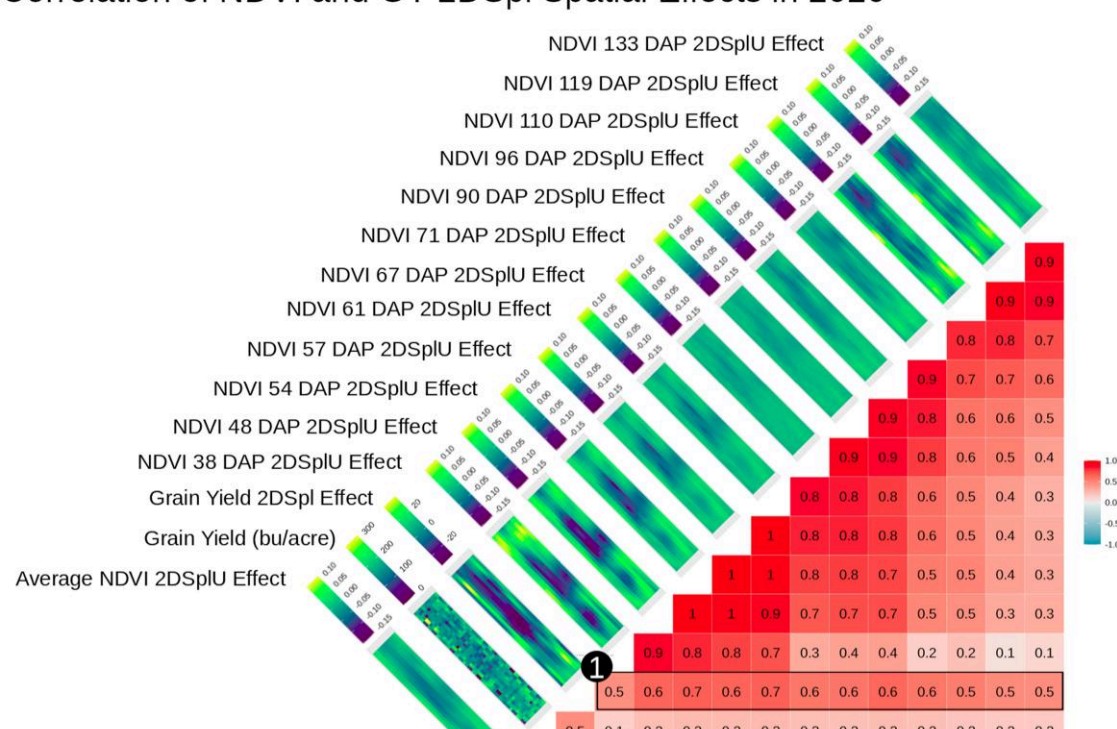

**Fig. 2.** 2020_NYH2 single timepoint 2DSplU NDVI PE estimates observed over 12 time points correlated with GY and GY 2DSpl spatial effects. Corresponding heatmaps showed values over the rows and columns of all experimental plots in the field and revealed similar spatial patterns. As indicated by (1), correlations of 0.5 to 0.7 between GY 2DSpl and the 2DSplU NDVI PE were found throughout the growing season, even early on at 38 DAP.

around 90 to 100 DAP, and in 2020 a gradual decline for GY and a gradual incline for GM. In contrast, there was a similar behavior between the GY and EH spatial effects in all years. Supplementary Fig. 12 illustrates the AR1 analog of Fig. 4 with agronomic trait AR1 spatial effects instead of 2DSpl effects and demonstrated weaker correlations to the NDVI PE in all traits and all years. Highly similar patterns in the correlation curves were observed; however, there was a tendency for the GM AR1 spatial effects to correlate with RRID, RR2DSpl, and RRAR1 NDVI PE more strongly than the 2DSpl or AR1 NDVI PE.

Simulation tested the efficacy of the first stage in detecting known environmental field effects. In each of the six simulation processes (linear, 1D-N, 2D-N, AR1xAR1, random, and RD) 10 iterations were performed, each time generating a new simulation. The simulated environmental variance was tested at 0.1, 0.2, and 0.3 times the proportion of phenotypic variation, and the correlation between time points was tested at 0.75, 0.90, and 1. Supplementary Figs. 22–24 illustrate the results for all simulation scenarios using 2017_NYH2, 2019_NYH2, and 2020_NYH2 NDVI phenotypes, respectively, demonstrating the impacts of varying the simulation environmental variance as well as the correlation of simulated environmental effects across the growing season. Increasing the variance tended to slightly increase the prediction accuracy, while decreasing the correlation between time points tended to decrease prediction accuracy. Figure 5 aggregates the prediction accuracies for the linear, 1D-N, 2D-N, AR1xAR1, and RD simulation scenarios and illustrated model groupings determined by a Tukey Honest Significant Difference (HSD) test.

The RREuc model showed relatively poor performance. This is possibly due to a mismatch in the geometry of the experiment because in reality the plots in the field were rectangular (e.g. 10 ft by 3 ft) and not perfectly square. The RREuc model was also most sensitive to the tested years and to changes in simulated variance and correlation. The AR1U model performed best in the AR1xAR1 scenario, while the 2DSplU model performed well in the Linear scenario. Specifying the PE covariance matrix allowed the RRAR1 and RR2DSpl models to perform consistently well in the 1D-N and 2D-N scenarios; however, by making no assumptions on the spatial structure the RRID model performs on average less than 10% worse and with comparable consistency.

## Estimation of spatial effects

First-stage NDVI PE were incorporated into the second stage of the proposed spatial correction approach using two distinct implementations, either modeling L or FE. Supplementary Fig. 13 illustrates genomic heritability, model fit, and GEER in the four years for GY, GM, and EH for all two-stage models when modeling L random effects. Baseline and spatially corrected GBLUP models, named G, G + 2DSpl, and G + AR1 representing Equations (5a), (5b), and (5c), respectively, are shown. Statistical significance of the spatial corrections and two-stage models were compared to the baseline G model using a paired $t$-test. The best models were determined by ranking the $t$-test $P$-value of GEER for GY. The best five two-stage L models, representing Equation (7), were G + L_AR1U, G + L_AR1M, G + L_2DSplU, G + L_2DSplM, and G + L_RRID.

Alternatively, modeling NDVI PE as FE followed four distinct definitions in Equations (8a–8d). Supplementary Fig. 17 illustrates genomic heritability, model fit, and GEER in the four years for GY, GM, and EH for all two-stage models when modeling FE. The

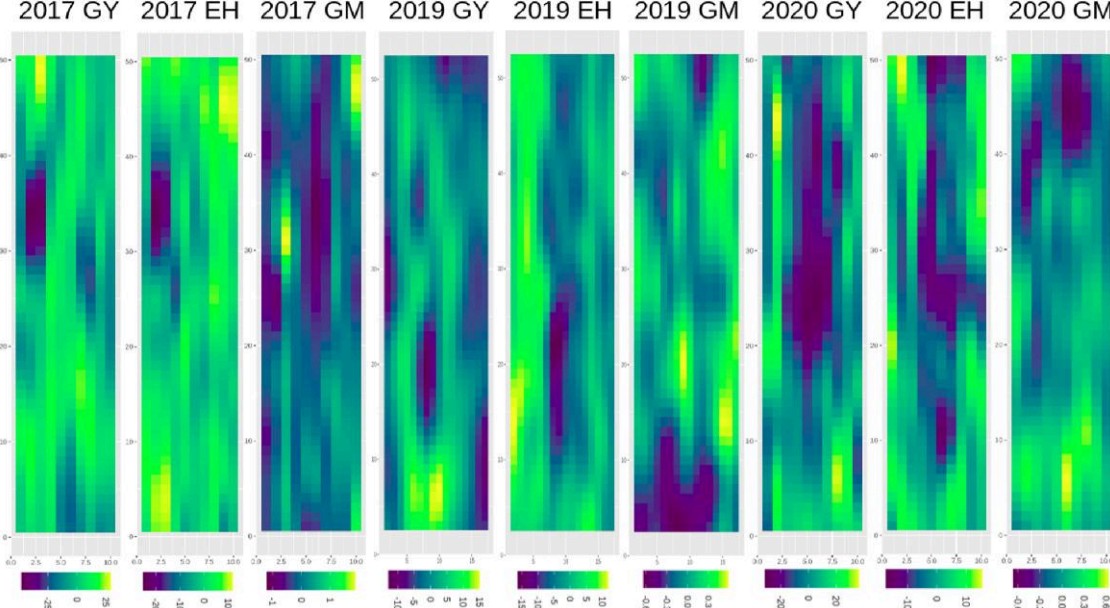

**Fig. 3.** 2019_NYH2 single timepoint 2DSplU NDVI PE estimates observed over six time points correlated with GY and GY 2DSpl spatial effects. Corresponding heatmaps showed values over the rows and columns of all experimental plots in the field and revealed similar spatial patterns. Illustrated are (1) correlations between GY 2DSpl and the 2DSplU NDVI PE, (2) correlations between soil EC and NDVI PE. and (3) correlations between soil d2Alt and NDVI PE. The AR1 analog is found in Supplementary Fig. 11.

**Table 4.** Correlations between average NDVI PE in 2019_NYH2 computed using the listed first-stage models and the soil EC, dEC, d2EC, Alt, dAlt, and d2Alt measurements.

| Model[a] | Soil EC | Soil dEC | Soil d2EC | Soil Alt | Soil dAlt | Soil d2Alt |
|---|---|---|---|---|---|---|
| 2DSplU | 0.46 | 0.21 | 0.04 | −0.06 | 0.11 | 0.21 |
| 2DSplM | 0.21 | 0.12 | 0.06 | 0.08 | 0.34 | 0.29 |
| AR1U | 0.39 | 0.18 | 0.04 | −0.06 | 0.06 | 0.16 |
| AR1M | 0.01 | 0.04 | 0.06 | 0.10 | 0.25 | 0.21 |
| RRID | 0.10 | 0.06 | 0.04 | 0.04 | 0.14 | 0.17 |
| RREuc | −0.06 | 0.03 | 0.09 | 0.07 | 0.14 | 0.15 |
| RRAR1 | −0.18 | −0.06 | 0.03 | 0.11 | 0.23 | 0.21 |
| RR2DSpl | −0.19 | −0.07 | 0.02 | 0.13 | 0.24 | 0.23 |
| RRSoilEC | −0.22 | −0.08 | 0.03 | 0.12 | 0.25 | 0.22 |
| RRSoilAlt | −0.21 | −0.09 | 0.04 | 0.12 | 0.25 | 0.23 |

[a]2DSplU, two-dimensional spline spatial model fit to single timepoints; 2DSplM, two-dimensional spline spatial model fit to multiple timepoints; AR1U, first-order autoregressive spatial model fit to single timepoints; AR1M, first-order autoregressive spatial model fit to multiple timepoints; RRID, random regression using an identity matrix to model plot-to-plot correlations for PE; RREuc, random regression using Euclidian distance to model plot-to-plot correlations for PE; RR2DSpl, random regression using 2DSpl spatial effect estimates to model plot-to-plot correlations for PE; RRAR1, random regression using first-order autoregressive spatial effect estimates to model plot-to-plot correlations for PE; RRSoilEC, random regression using soil electrical conductance measurements to model plot-to-plot correlations for PE; RRSoilAlt, random regression using altitude measurements to model plot-to-plot correlations for PE.

baseline and spatially corrected GBLUP models, named G, G + 2DSpl, and G + AR1, respectively, are shown. As before, the best two-stage FE models were determined by ranking the *t*-test *P*-value of GEER for GY. The top eight models were named G + Havg_AR1U, G + Havg_2DSplU, G + Havg_2DSplM, G + Favg_2DSplU, G + H3_2DSplM, G + H3_RRID, G + F3_AR1U, and G + F3_2DSplU.

To measure performance of the proposed two-stage approach over the baseline G model, a difference (G Diff) was computed within each of the four years for heritability, model fit, and GEER. In the following analyses, the baseline G and G + 2DSpl models include results from 2015, 2017, 2019, and 2020, while the baseline G + AR1 model excludes 2015 due to AR1 convergence issues. Figure 6 illustrates G Diff for the baseline G + 2DSpl and G + AR1 spatial correction models and for the best six models defining L (G + L) and FE (G + H). Supplementary Figs. 14 and 18 illustrate all two-stage models when defining L and FE, respectively. The spatially corrected baseline models, G + 2DSpl and G + AR1, demonstrate improvements over G. The G + 2DSpl model provided significant improvements in heritability and model fit for all traits, and a significant improvement in GEER for GY; while, G + AR1 provided significant improvements in model fit for all traits, and a significant improvement in heritability and GEER for GY.

Figure 6 demonstrates further improvements for the two-stage models over the baseline G, G + 2DSpl, and G + AR1 models. Two-stage models incorporating NDVI PE improved heritability and GEER for GY, GM, and EH more than the baseline spatially corrected models. The best two-stage models translated increased heritability to an increase in GEER. Improvements to GY GEER over baseline G, G + 2DSpl, and G + AR1 models were summarized in Table 5 for the six best two-stage models when incorporating NDVI PE. Table 5 columns of "ΔG", "ΔG + 2Dspl", and "ΔG + AR1" indicate the mean and standard deviations of model differences in GEER for GY for the four years compared to G (G Diff), G + 2DSpl (G + 2DSpl Diff), and G + AR1 (G + AR1 Diff), respectively. Supplementary Figs. 15 and 19 illustrate the G + 2DSpl Diff for all models when defining L and FE, respectively. Supplementary Figs. 16 and 20 illustrate the G + AR1 Diff for all models when defining L and FE, respectively. While GEER for GY and EH was improved, none of the FE models significantly improved GEER for GM, a result potentially attributable to the lower correlations between NDVI PE and GM spatial effects seen in Fig. 4 and Supplementary Fig. 12, the difficulty in detecting GM

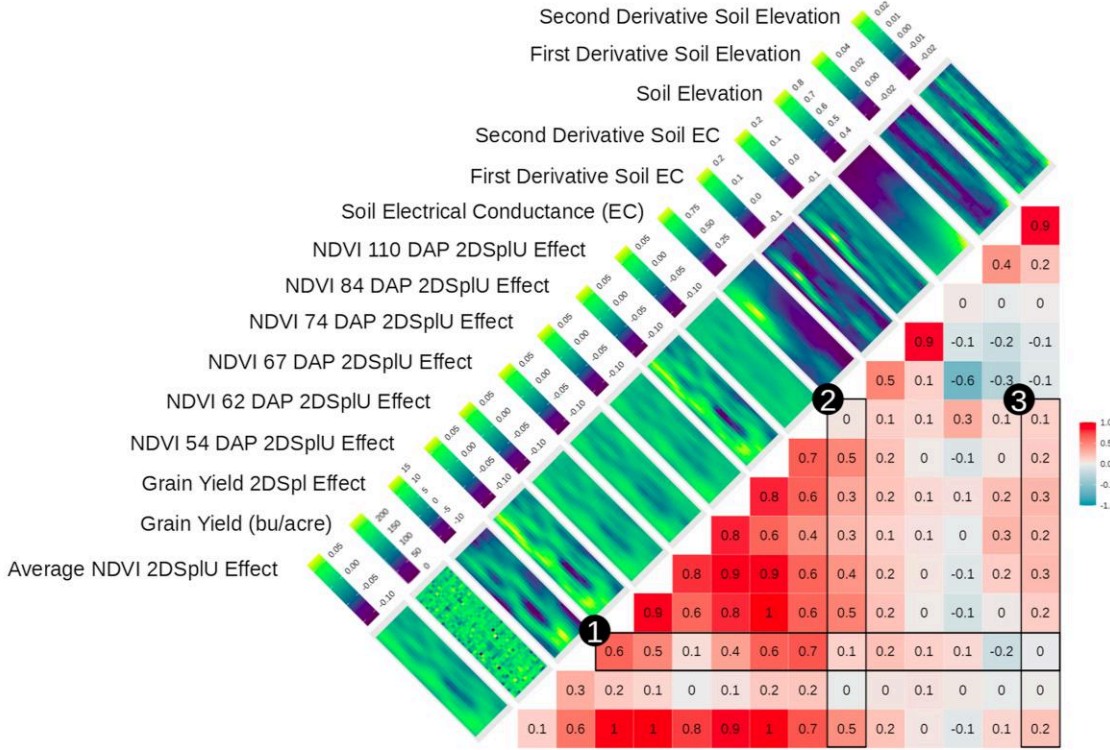

**Fig. 4.** Correlations between 2DSpl spatial effects of agronomic traits and PE from the tested first-stage models. Spatial effects for GY, GM, and EH in the 2015_NYH2, 2017_NYH2, 2019_NYH2, and 2020_NYH2 field experiments were compared with model PE across the growing season. RR models including soil information rather than NDVI in the first-stage, named RRSoilAlt and RRSoilEC, were also included for 2019_NYH2. The AR1 analog is found in Supplementary Fig. 12.

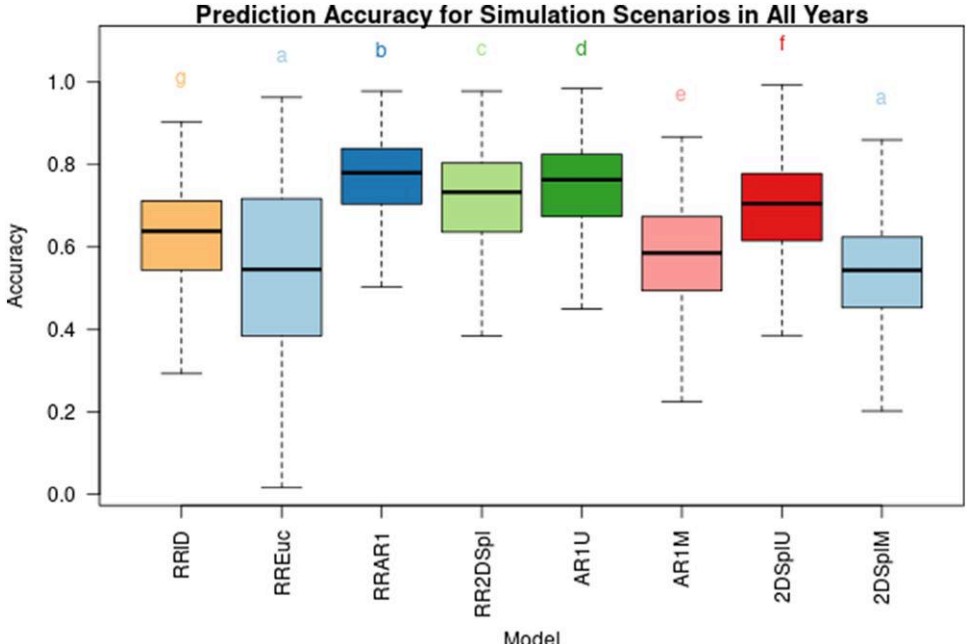

**Fig. 5.** Aggregated prediction accuracy of the tested first-stage models for the linear, 1D-N, 2D-N, AR1xAR1, and RD simulation scenarios using the 2017_NYH2, 2019_NYH2, and 2020_NYH2 NDVI data. Prediction accuracy is the correlation of the simulated environmental effect and the model's recovered environmental effect. Models are grouped together after performing a Tukey's HSD test.

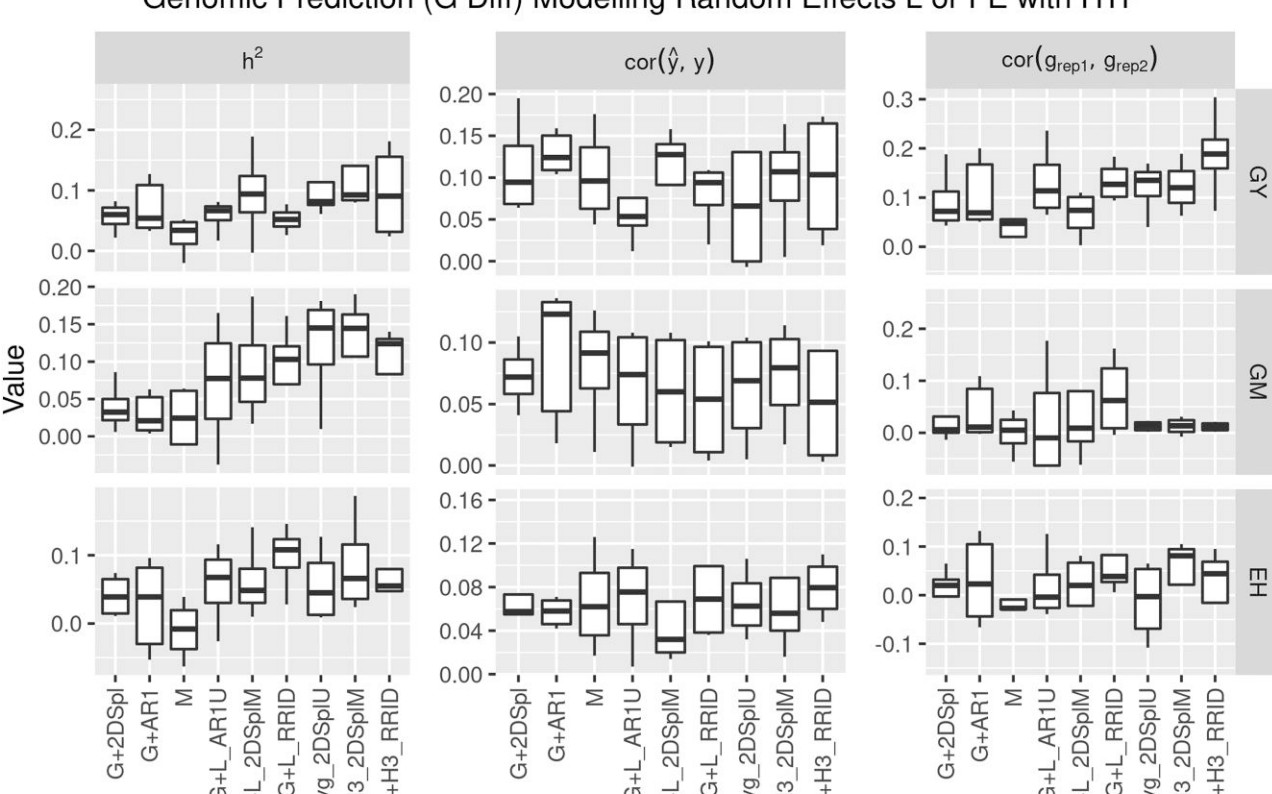

**Fig. 6.** Differences compared to G (G Diff) for genomic heritability, model fit, and GEER for GY, GM, and EH in the four years. The models G, G + 2DSpl, and G + AR1 were baseline GBLUP and spatially corrected GBLUP models using 2DSpl and first-order autoregressive spatial models, respectively. M is a baseline multitrait model. Illustrated are the best three two-stage models using L (G + L) and the best three two-stage models using FE (G + H), determined by ranking *t*-test *P*-value of GEER for GY. The G + L and G + H models have L and FE, respectively, defined using NDVI PE of corresponding names.

**Table 5.** The best two-stage models vs the baseline GBLUP models (G, G + 2DSpl, G + AR1) when comparing the correlation of GEER for GY.

| Model[a] | ΔG | ΔG + 2DSpl | ΔG + AR1 |
|---|---|---|---|
| G + H3_RRID | 0.188 ± 0.094 (*) | 0.095 ± 0.045 (*) | 0.082 ± 0.053 (+) |
| G + H3_2DSplM | 0.123 ± 0.055 (*) | 0.029 ± 0.074 | 0.025 ± 0.09 |
| G + Havg_2DSplU | 0.12 ± 0.056 (*) | 0.026 ± 0.048 | 0.012 ± 0.057 |
| G + L_2DSplM | 0.065 ± 0.049 (*) | −0.028 ± 0.105 | −0.056 ± 0.123 |
| G + L_AR1U | 0.132 ± 0.077 (*) | 0.038 ± 0.035 (+) | 0.041 ± 0.031 (+) |
| G + L_RRID | 0.133 ± 0.041 (*) | 0.039 ± 0.043 (+) | 0.036 ± 0.049 |

The symbols (*) and (+) denote *t*-test *P*-values less than 0.05 and 0.1, respectively.
[a]H3, second-stage spatial corrections modeled as fixed regressions on first-stage spatial effect estimates from three distinct growth phases; Havg, second-stage spatial corrections modeled as a fixed regression on the average first-stage spatial effect estimates; L, second-stage spatial corrections modeled using a plot-to-plot correlation matrix calculated using first-stage spatial effect estimates; RRID, first-stage random regression model using an identity matrix to model plot-to-plot correlations; 2DSplM, first-stage, multitimepoint model using 2DSpl to estimate spatial effects; 2DSplU, first-stage, single time point model using 2DSpl to estimate spatial effects; AR1U, first-stage, single time point model using first-order autoregressive correlations to estimate spatial effects.

spatial effects seen in Table 3, and the small GM spatial variation seen in Fig. 1.

Drawing from the simulation results in Fig. 5, the 2DSplM and AR1M models may have had less overall accuracy due to restricted estimation of spatial covariance components between time points, and thereby negatively impacted the simulation when weaker correlations (< 0.9) across time points were used. In real data, as seen in Figs. 2, 3, Supplementary Figs. 8, 9, and 10, NDVI PE tended to be strongly correlated (>0.8) between time points; this may have explained the improvements seen in Table 5 when the 2DSplM NDVI PE were incorporated into second-stage genomic prediction. The RRAR1 and RR2DSpl models

performed well in first-stage simulation; however, the second-stage genomic prediction was not particularly improved by these models. The RRID, AR1U, and 2DSplU models performed well in first-stage simulation and significantly improved the second-stage model accuracy, indicating these models provided robust detection of spatial heterogeneity.

The second stage in the proposed approach could use soil data as an alternative to NDVI PE from the first-stage. Figure 7 illustrates differences in 2019_NYH2 against the baseline G (G Diff) for genomic heritability, model fit, and GEER for GY, GM, and EH. The best eight models when modeling L (G + L) or FE (G + H) using soil data are illustrated in Fig. 6; however, Supplementary

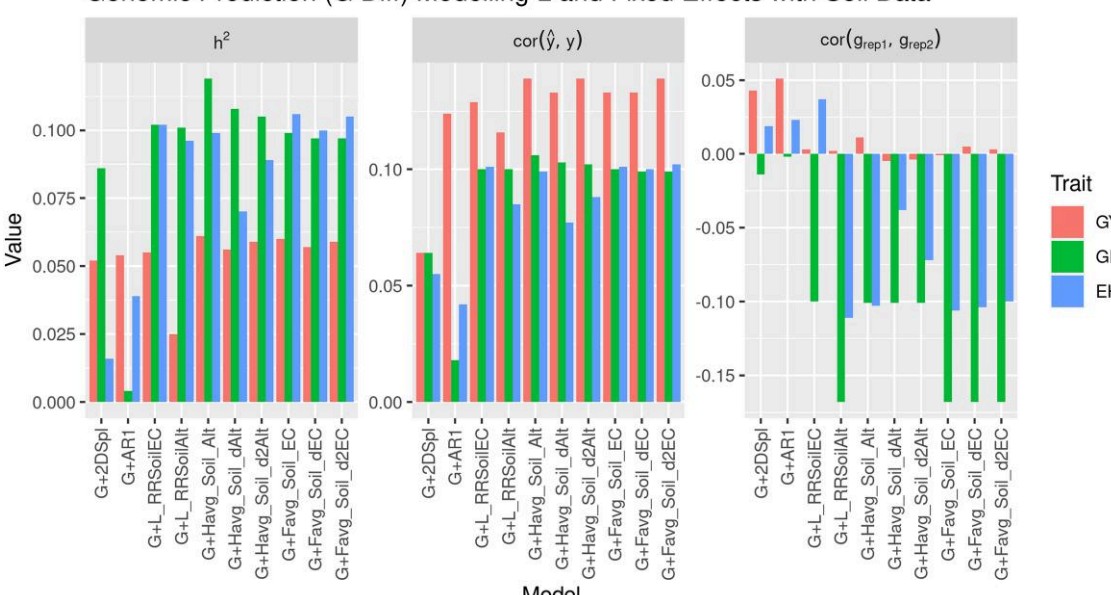

**Fig. 7.** Differences compared to G (G Diff) for genomic heritability, model fit, and GEER in 2019_NYH2 for GY, GM, and EH, with soil data implemented as a plot-to-plot correlation matrix (g + l) or as fixed effects (g + h). The best eight models using soil altitude (Alt), soil electrical conductance (EC), and the first and second derivatives (dAlt, d2Alt, dEC, d2EC) were illustrated.

Fig. 21 illustrates all of the models using soil data. Again, the baseline spatially corrected models, G + 2DSpl and G + AR1, are shown. Included were the L models named G + L_RRSoilEC and G + L_RRSoilAlt, and the FE models named G + Favg_Soil_Alt, G + Favg_Soil_dAlt, G + Favg_Soil_d2Alt, G + Havg_Soil_EC, G + Havg_Soil_dEC, and G + Havg_Soil_d2EC. The soil information increased GM and EH heritability, and model fit for all traits, more than the baseline G + 2DSpl and G + AR1 models; however, for all traits GEER performed lower than the baseline G + 2DSpl and G + AR1 models, particularly for GM and EH.

This approach to incorporate soil data did not improve model genotypic effect accuracy for GY; however, this result was from a single field experiment in a single year. Similar to how the NDVI HTP itself did not correlate highly with GY while the spatial effects of NDVI correlated strongly with the spatial effects of GY, the spatial effects of the soil data may prove more beneficial for improving prediction accuracy than the soil data itself. The soil data had much weaker correlations than the NDVI to agronomic traits, with a high of 0.07, 0.03, and 0.20 for GY, GM, and EH, respectively, compared to NDVI with a high of 0.67, 0.60, and 0.46 for GY, GM, and EH, respectively. Further indicating persistent spatial effects may be limiting the effectiveness of soil EC data in this study, Table 4 illustrates that including the soil data into RR models resulted in NDVI PE relatively well correlated with the soil elevation gradients, but negatively correlated with the soil EC data itself. Furthermore, soil data may need to be incorporated with weather information in order to effectively estimate the benefit or detriment of the local environmental effect. For instance, low elevation can be either beneficial or detrimental depending on rainfall.

## Conclusion

In all years and for all agronomic traits, correlations between the agronomic trait spatial effects and NDVI PE were higher than correlations between the agronomic traits and NDVI themselves, indicating the spatial patterns in NDVI do provide information on spatial patterns observed for key agronomic traits. Furthermore, the NDVI PE from 2DSpl, AR1, and RR models consistently identified the same poorly performing regions in the field over the growing season, and identified substantially the same regions as the baseline GY and EH spatial effects. Baseline GM spatial effects showed an inverted behavior with NDVI PE and were less localized than for GY and EH. The soil EC correlated most with NDVI PE from the 2DSplU and AR1U models across time. Therefore, spatial heterogeneity quantified by NDVI PE corresponded strongly with agronomic trait spatial effects and soil EC.

Incorporating first-stage NDVI PE into the second-stage spatial corrections for GY, EH, and GM either as a covariance of random effects (L) or as FE, significantly improved heritability, model fit, and GEER. In simulation, the RRAR1 and RR2DSpl models performed strongly; however, only the RRID, AR1U, and 2DSplU models performed well in simulation and also improved the two-stage spatial correction for agronomic traits. The RRID model performed consistently above average in simulation and improved spatial correction performance of GY and EH experimentally. Notably, the RRID model made no spatial assumptions, which may facilitate deployment compared to models which rely on spatial annotations. The equilibrium between model generalizability and model over-specification when detecting NDVI PE was balanced most by the RRID, AR1U, and 2DSplU models.

Aerial image HTP provided greater understanding of spatial heterogeneity in the field, and when coupled into the proposed two-stage spatial correction approach, enabled a more effective spatial correction than any of the baseline models (G + 2DSpl, G + AR1, and M). Furthermore, the observed spatial heterogeneity could be partially explained using soil EC and elevation. Further research is needed to identify more informative image features and develop novel statistical approaches for integrating HTP across the growing season with end-of-season agronomic trait prediction. To these ends, larger datasets are required to evaluate the proposed approaches, and the continued aggregation of FAIR data is crucial.

## Data availability

This study used phenotypic data of hybrid maize (Z. mays L.) field experiments part of the G2F program planted in 2015

(https://doi.org/10.25739/erxg-yn49), 2017 (https://doi.org/10.25739/w560-2114), 2019 (https://doi.org/10.25739/t651-yy97), and 2020 (https://doi.org/10.25739/hzzs-a865), named 2015_NYH2, 2017_NYH2, 2019_NYH2, and 2020_NYH2, respectively. The genotypic SNP marker data was also from the G2F program (https://doi.org/10.25739/frmv-wj25). The collected image data from 2015, 2017, 2019, and 2020 are available in the Supplemental section of this manuscript.

Supplemental material available at GENETICS online.

## Acknowledgments

The authors would like to thank the G2F consortium for providing the NYH2 field experiments from 2015 to 2020 used in this study. This consortium involves more than 30 researchers representing more than 20 research institutions. Details about the initiative and publicly available resources can be found at www.Genomes2Fields.org. Thanks to Chris Hernandez, Peter Selby, Sam Bouabane, Ranjita Thapa, Simon Reinhard, Lynn Johnson, Seth Murray, Jacob Washburn, Filipe I. Matias, Annarita Marrano, and Felipe Sabadin for their help and suggestions on the image processing pipeline and the research more broadly.

## Funding

This work was supported by the U.S. Department of Agriculture National Institute of Food and Agriculture, Hatch projects 1024080 (K.R.R.), 100397 (M.A.G), 1010428 (M.A.G.), 1013637 (M.A.G.). 1013641 (M.A.G.), Iowa Corn Promotion Board, and Cornell University startup funds (K.R.R. and M.A.G.).

## Conflicts of interest

The author(s) declare no conflict of interest.

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

*Editor: M. Sillanpää*