## [Peer Review File · Genetics]

Spatio-temporal modeling of high-throughput multi-spectral aerial images improves agronomic trait genomic prediction in hybrid maize

Nicolas Morales, Mahlet Anche, Nicholas Kaczmar, Nicholas Lepak, Pengzun Ni, M. Romay, Nicholas Santantonio, Edward Buckler, Michael Gore, Lukas Mueller, and Kelly Robbins

NOTE: The reviews and decision letters are unedited and appear as submitted by the reviewers.

In extremely rare instances and as determined by a Senior Editor or the EIC, portions of a review may be redacted. If a review is signed, the reviewer has agreed to no longer remain anonymous.

The review history appears in chronological order.

Review Timeline:

Submission Date:	2022-10-17
Editorial Decision:	2022-12-05
Resubmission Received:	2023-12-02
Editorial Decision:	2024-01-26
Resubmission Received:	2024-02-07
Accepted:	2024-02-18

December 5, 2022

GENETICS-2022-305661

Spatio-temporal modeling of high-throughput multi-spectral aerial images improves agronomic trait genomic prediction in hybrid maize

Dear Dr. Morales:

Two experts in the field have reviewed your manuscript, and I have read it as well.

The work may become important contribution to the field. While your manuscript is not currently acceptable for publication in GENETICS, we would welcome a substantially revised manuscript. Both reviewers have comments and concerns to be addressed in a revised manuscript. You can read their reviews at the end of this email.

I think all the reviewer comments are very much needed to make your paper easier to read. Please try to add all details of the statistical models and correlation structures better available for readers. I think all the reviewer comments will clarify your paper and improves overall readability which will be very good in the end. We look forward to receiving your revised manuscript. Please let the editorial office know approximately how long you expect to need for revisions.

Upon resubmission, please include:

1. A clean version of your manuscript;
2. A marked version of your manuscript in which you highlight significant revisions carried out in response to the major points raised by the editor/reviewers (track changes is acceptable if preferred);
3. A detailed response to the editor's/reviewers' feedback and to the concerns listed above. Please reference line numbers in this response to aid the editor and reviewers.

Your paper will likely be sent back out for review.

Additionally, please ensure that your resubmission is formatted for GENETICS
<https://academic.oup.com/genetics/pages/general-instructions>

Follow this link to submit the revised manuscript: <https://genetics.msubmit.net/cgi-bin/main.plex?el=A5NR3FCc5A5ThM7I5A9ftdpreVJJMIY50nzmXWrNHsgZ>

Sincerely,

Mikko J. Sillanpää

Associate Editor

GENETICS

Approved by:

Sharon Browning

Senior Editor

GENETICS

Reviewer #1 (Comments for the Authors (Required)):

This paper considers modelling of a spatio-temporal dataset obtained by high-throughput phenotyping of replicated field trials with hybrid maize. The focus is on NDVI as assessed over the growing season and correlated agronomic traits. Analysis is based on various

models that account, in different ways, for spatial, temporal and trait-to-trait correlation. The ultimate objective is to improve analysis for targeted agronomic traits, which are correlated with the NDVI data. The approach taken by the authors is described as a two-stage approach.

(1) The authors consider four experiments. The design of these experiments is not described at all. It is important to provide information on the field layout, the randomization and blocking structure and the number of replications and genotypes. The reason this is important is because any analysis should account for the experimental design. It appears that the design is largely ignored. For example, I would expect there to be replicates and possibly incomplete blocks, assuming that an efficient design was used. Such design effects would need to be included in at least a baseline analysis. For readers to convince themselves to what extent this has been done, a succinct description of the design is needed, including any blocking structure. Incidentally, the introduction asserts that randomization can help control errors. It may be argued that randomization only helps assessing and accounting for errors, but it does not really control errors. The key design feature that allows error control is blocking.

(2) For the data at hand, correlation needs to be taken into account in three dimensions: (i) spatial, (ii) temporal and (iii) trait-to-trait. The different model approaches do this to varying degrees, and it appears that none or not all of the models accounts for all sources of correlation simultaneously, which does not seem satisfactory. I will come back to this general point when considering specific models.

(3) The outline of the models suffers from an inconsistency in notation across models. The description at times gives the impression of being extracted from manuals and papers related to different packages used in the various an analysis, without any serious attempt made to unify the notation. This makes comparison of models difficult for a reader. Also, the description of model terms is often incomplete or inaccurate. This general point will also be taken up when discussing specific models. There is a simple litmus test for a sufficient model description: If the reader is equipped with my data and my description of the methods in my M&M section, would he or she be in a position to reproduce my results? I am afraid to say the answer to this question for the present submission is a clear 'no!'.

(4) The authors use a two-stage approach to fitting their model. Different readers may expect different things under the general label "two-stage approach." I had to read the paper twice before starting to fully grasp what the two stages are. It would be useful early on, before a detailed description of any models, to explain in general terms what exactly is done in the first stage and what is done in the second stage of the analysis.

(5) I had some trouble understanding what the authors mean by the term "local environmental effect (LEE)". This term sounds a bit cryptic. It would be much easier to understand if this term were introduced via a linear model being fitted. This could have an explicit effect for what the "LEE." My guess is that what the authors mean is simply the non-genetic effects associated with an observation / plot, i.e. possibly time-varying environmental effects for plots, blocks, replicates, as opposed to the genotypic effects of interest. But a clarification, best via a statistical model and when the term is first introduced, would be very useful.

(6) Equation 1: The residual "e" needs to be explained. What are the fixed effects in beta? What is a "local environment"? Isn't this just a plot, and is this why u_p carries the subscript "p"? Is this a model for a single time point or for multiple time points? Why does this model have no effects representing the randomization layout of the replicated field experiment (replicates, blocks)?

(7) In the text following Equation 1, and also that preceding Equation 3, the authors refer to the multivariate case in the case where multiple time points are considered for the same trait. I find the term "multivariate" misleading in this context, where the correlation that needs to be accounted for is one for serial or autocorrelation. Also, as the same trait is being measured at each time point, heterogeneity of variance is not necessarily an issue, as it would be with multiple traits. So perhaps the authors can reconsider their terminology here, also because their analysis ultimately focuses on the correlation between NDVI and agronomic traits of interest, which is what I would consider as a multivariate (multitrait) problem.

(8) Equation 4: It is not clear how this scalar equation ties in with the preceding models stated in matrix form. This is also because the notation looks completely different.

(9) Equation 5: The text purports that this equation give a structure for spatial correlation according to the 2DSpl model, but the structure clearly implies independence between all plots.

(10) Equation 6: The text here suggests that the model is for the "multivariate case", i.e. for the different time points, but the structure for Σ_u is diagonal, implying independence between time points, which is not a realistic model for repeated measurements.

(11) Equation 8: Same problem as with Equation 6.

(12) Equation 9: This equation is for a further approach to spatio-temporal modelling. However, comparison with previously described models is made difficult by yet another change in notation. This equation is scalar. The model is announced to be for repeated measurements, but neither of the two random effects a_{kl} and p_{bl} is indexed by time. The only time-varying effects seems to be the residual $e_{ikn:t}$, so it is not clear how this model can account for serial correlation. Nor is it clear how this model accounts for spatial correlation. The description of the incidence matrices Z_a and Z_p would need to be given in detail for readers to appreciate how spatial and temporal correlation are accounted for. The authors just mention on page 11 that Legendre polynomials or linear spline functions can be used, but this is not enough. What exactly did the authors use as covariates? Certainly, there would need to be time indices involved in the scalar expressions in Equation 9 for this model to account for both spatial and temporal correlation. The matrix expressions given in the first paragraph on page 11 do involve subscripts for time, but the notation is not in agreement with Equation 9. So something is wrong here. The subscript for time itself has subscripts i and j , which is also confusing, because the subscript i has a different meaning in Equation 9. The fixed effect F_i is described as effect for the "replicate nested with image event data", but the effect is not indexed by t , which is at odds with this description. It is good, though, to have design effects in the model. Why is this not done for the other models as well? Later in the text, a matrix expression is given for the variance-covariance matrix of the response vector y . For this to be comprehensible, the ordering of scalar observations in the vector y would need to be given.

(13) The description of the structures for E on the lower part of page 11 is not succinct. Only one explicit equation is given for E_{ij} . For this second option, isotropy seems to be assumed. Is this a realistic assumption? What exactly are "positions"? The third to fifth option is cryptic. What are "empirical correlations"? How exactly are they computed? Explicit equations would help here. What is the underlying rationale?

(14) The description of the simulation on page 12 is very scant. It remains unclear from what model exactly the data are simulated. The same goes for the evaluation of accuracy, which is denoted as a five-step process. It mentions a first-stage model, but it is unclear what exactly a first-stage model is here. Also see my comment (4). In a second step, "the computed NLEE were subtracted from NDVI HTP in order to minimize latent spatio-temporal effects in the NDVI." This subtraction is never mentioned up to here, also not with the description of the various models. This would seem to be a key step in the two-stage approach announced in the introduction. Should this not be described as part of the modelling approach? Also, the subtraction will induce correlations among the corrected values. How are these taken into account in the downstream analysis? Ignoring them will lead to invalid inferences. For an illustration based on a simple case see

Piepho, H.P., Williams, E.R., Ogutu, J.O. (2013): A two-stage approach to recovery of inter-block information and shrinkage of block effect estimates. *Communications in Biometry and Crop Science* 8, 10-22.

(15) On page 13, the stage-wise approach is mentioned again. Specifically, a "first-stage NLEE" is mentioned. What exactly is this? It seems to be some kind of estimator derived from the models fitted to the NDVI data, but it seems this is not really explained anywhere. The subsequent Equation 13 used the empirical correlation among the NLEE over time to estimate the matrix L. An explicit equation for the correlation would be useful here. For the second option, based on "fixed factor effects (FE)", the key feature of the models seems to be a regression of the agronomic response on the NLEE as covariates. The description of the regression terms is a bit cryptic because it is not made explicit what forms the H and F matrices take and how NLEE features in these.

(16) Model performance is assessed via the heritability. The equation given seems to be on a plot basis. This should be made explicit and also justified. In field trials, heritability is

usually assessed on a plot basis. Also, I am not sure the genetic variance can just be plugged in as done here because it is estimated from a model involving the GRM. How does this simple equation for heritability account for the pairwise genetic covariance among hybrids? And how does it account for the environmental covariance among residual errors "e"?

(17) Table 3: It is not clear how the correlations reported in this table were estimated. Is this based on a trivariate model for the three agronomic traits? It seems that the section "Agronomic Genomic Prediction" only describes univariate (single-trait) models. Even the "multi-trait model" in Equation 12 is only for one agronomic trait at a time.

Reviewer #2 (Comments for the Authors (Required)):

In general, I liked the manuscript. It brings up an interesting topic and presents an array of modelling approaches to capture spatial heterogeneity in phenotypic variation in a secondary trait (vegetation index) and link that to spatial heterogeneity in target agronomic traits. I think the work can be very relevant as many research platforms are putting their trust in high-throughput phenotyping methods.

Having said that, I have some points of concern which I will try to explain below and which, as it happens, I believe become progressively more important.

One point that is missing is a discussion about the validity of the aerial images themselves as a proxy for spatial heterogeneity in local biomass/'green-up' effects. A UAV flying at a particular date may/does not differentiate true spatial heterogeneity from e.g.

heterogeneity in phenological stages of different plants. Therefore what you see as spatial variation in the field may partly be variation in phenology rather than 'local spatial effects'. Not sure how relevant this problem is for prediction since you find local environment effects from the images (NLEE) to correlate with spatial heterogeneity in grain yield, but, judging from Fig. 2, correlations are not that high and NLEE appears to correlate weakly

with soil parameters (Table 4). I therefore wonder to what extent we're looking at true spatial effects vs. different plant development stages. I guess it's an inherent caveat to HTP methodologies in general, but it would be worth discussing as this paper heavily revolves around spatial heterogeneity in the field.

On the subject of spatial heterogeneity in the field, it is hard to judge from the text and figures how important spatial effects are in the studied trials. We are presented with spatial images of the field without the raw, phenotypic variation in the traits of interest as a reference. Also, if I'm correct the experimental design has never been mentioned; all I know is that there are two replicates (mentioned on p.15). The point is, in my experience working with spatial models in many a trial, most spatial variation is captured by the experimental design (e.g. by adding row/column effects to the model) and what spatial heterogeneity the spatial model picks up is usually quite small in relation to raw phenotypic variation. To put the importance of NLEEs into perspective I think the reader needs to be given (visual) guidelines as to the importance of these NLEEs in general.

A more general point about the manuscript is that it's quite dense, with many models run and many results presented. I wonder if the manuscript could be tidied up a bit. For example, by adding a table that lists all models with their distinct random and fixed effects structures and a conceptual figure that shows how the first and second stage of the analyses are linked. Also, I wonder if the number of models can be trimmed down a bit - or at least better justified/introduced. For example, random regression models (RRs) are invoked as an alternative to running spatial models at each time point. This seems like an appealing idea but could work only, in my view, if the curves can be fitted accurately. We don't get an impression reading the manuscript of how well the curves fit the data (3rd-order polynomials are used). If they don't fit well, we would a priori not expect the RRs to accurately predict spatial variation in the target trait, which is indeed what we see for grain yield in Figs. 4 and S8.

A final point is that the authors should try to get the overall aim of the manuscript - or its value - as well as its conclusion more clear. The authors aim to investigate whether "local environment effects are able to detect spatial heterogeneity in agronomic traits" and whether these effects can aid in the genomic prediction. I'd say the latter is the most interesting part for breeders as we're ultimately interested if we can get accurate predictions, yet the manuscript puts in my view too much emphasis on 'getting the right spatial model'. We've already established that the correlation between local env. effects and spatial heterogeneity in target traits is not very strong; what really matters for breeders, I think, is how plot-level (or genotype-level, rather) values for intermediate phenotypes relate to values of the target trait. That comes in the second-stage analysis, but to my

surprise there has been no cross validation. The authors basically looked at goodness of fit. For a true validation of your analyses you'd want to see how well your models do when predicting the phenotypes of 'unknown' genotypes. I think the authors need be clearer on the value of their analysis in this regard. This becomes also apparent in the conclusion section, which came across to me as somewhat inconclusive. This section mentions a lot of the model abbreviations and how they compared, but what is lacking a clear message, e.g. how we should go about leveraging HTP and spatial information to gain clearer predictions of our traits of interest.

My apologies for this long-winded review. I do like the ideas presented here and I like the variety of ways in which spatial effects are estimated; at this point, however, I think the manuscript is a bit too dense and lacking a clear goal. If the authors were somehow able to fix this it would make the manuscript much more interesting and relevant.

Some comments by section/page:

In general, please use subsections, especially in the methods section, to make it easier to navigate between the different analyses.

p.2, 2nd par: "...allow estimation of permanent environment effects from RR...". I think you need to introduce this concept a bit more. In animal breeding, a PE effect would indicate an effect of the animal on the phenotype independent of its genetics (e.g. food conditions during upbringing, body size, etc). Are you referring to the fact that individual plants (replicates) may consistently perform better or worse across time than other replicates from the same genotype? So if I'm correct, rather than estimating spatial effects across plots in the field at each time point you're estimating spatial heterogeneity as different longitudinal trajectories differing between individual plants/plots by fitting lines through time; you then look at the variation between longitudinal predictions at each time point (as per info on p11). I think a conceptual figure distinguishing the two approaches may be useful. But see my earlier comment on RR.

P4, Field Experiments: what is the design of the experiment? Later somewhere you mention there are two replicates.

P8, Eqn. 1: Please already give a brief explanation of the $Z_p u_p$ term. Also, how is the experimental design of the experiment factored into the model?

P10, eqn. 9: See my comment above about the use of RR.

P11, mid: what does n stand for in the equation?

P11, same par: "...third order Legendre polynomials were considered". How well did these polynomials capture the spatial trend in HTPs? Would longitudinal spline curves not make more sense in these types of data?

P13, eqns 11a-c: is there any other spatial correction to these models, based on experimental design?

P14, table 2. Please make reference to the equations; it's at times hard to remember which model represents which.

Reviewer 1:

This paper considers modelling of a spatio-temporal dataset obtained by high-throughput phenotyping of replicated field trials with hybrid maize. The focus is on NDVI as assessed over the growing season and correlated agronomic traits. Analysis is based on various models that account, in different ways, for spatial, temporal and trait-to-trait correlation. The ultimate objective is to improve analysis for targeted agronomic traits, which are correlated with the NDVI data. The approach taken by the authors is described as a two-stage approach.

Comment 1. The authors consider four experiments. The design of these experiments is not described at all. It is important to provide information on the field layout, the randomization and blocking structure and the number of replications and genotypes. The reason this is important is because any analysis should account for the experimental design. It appears that the design is largely ignored. For example, I would expect there to be replicates and possibly incomplete blocks, assuming that an efficient design was used. Such design effects would need to be included in at least a baseline analysis. For readers to convince themselves to what extent this has been done, a succinct description of the design is needed, including any blocking structure. Incidentally, the introduction asserts that randomization can help control errors. It may be argued that randomization only helps assessing and accounting for errors, but it does not really control errors. The key design feature that allows error control is blocking.

Author: In order to clarify this point, such design information was included in the model, and we amended the text to describe the design type, the number of genotypes tested, and the number of replicates and plots, for the four experiments (Line 160-170). The rows and columns were included in the models and are now described (Line 170-175). We added more detail in the model specification for the fixed effects, and added details about the tested germplasm. To clarify the introduction, we changed "control for errors" to "account for confounding genetic, spatial, and environmental variation" (Line 75).

Comment 2. For the data at hand, correlation needs to be taken into account in three dimensions: (i) spatial, (ii) temporal and (iii) trait-to-trait. The different model approaches do this to varying degrees, and it appears that none or not all of the models accounts for all sources of correlation simultaneously, which does not seem satisfactory. I will come back to this general point when considering specific models.

Author: We agree that there are three aspects to consider (i) spatial, (ii) temporal, and (iii) trait-to-trait. This study explores spatial and temporal information derived from high-throughput phenotype (HTPs) (in Equation 1) and agronomic traits (previously in Equations 11a-c and now in Equations 5a-c) independently. However, the trait-to-trait aspect is explored in two ways, 1) between agronomic and HTP traits using multi-trait models (previously in Equation 12 and now in Equation 6) and 2) between HTP time points (Equation 3). Combining the three aspects is explored by leveraging spatial and

temporal PE derived from HTP into genomic prediction in the two-stage models (previously in Equations 14a-d and now in Equations 8a-d). Ideally this would be done via a parsimonious single-stage model; however, the models and results as presented demonstrate the relationship between spatial variation in HTP and agronomic traits and the benefits of using longitudinal HTP to improve the accuracy of spatial corrections in agronomic traits. We will continue to pursue improved models moving forward, but we feel these results represent a significant contribution to the literature and should be shared with the community to spur further research on this topic.

Comment 3. The outline of the models suffers from an inconsistency in notation across models. The description at times gives the impression of being extracted from manuals and papers related to different packages used in the various an analysis, without any serious attempt made to unify the notation. This makes comparison of models difficult for a reader. Also, the description of model terms is often incomplete or inaccurate. This general point will also be taken up when discussing specific models. There is a simple litmus test for a sufficient model description: If the reader is equipped with my data and my description of the methods in my M&M section, would he or she be in a position to reproduce my results? I am afraid to say the answer to this question for the present submission is a clear 'no!'.

Author: We appreciate the concern and have made an effort to unify the notation, ensuring that matrices are bold and uppercase throughout. The models also now consistently reference the design effects modeled as fixed effects (Line 295, 440, 465, 480, 500). Model numbers were reordered to be more concise. The main text now only uses matrix notation for the numbered equations.

Comment 4. The authors use a two-stage approach to fitting their model. Different readers may expect different things under the general label "two-stage approach." I had to read the paper twice before starting to fully grasp what the two stages are. It would be useful early on, before and detailed description of any models, to explain in general terms what exactly is done in the first stage and what is done in the second stage of the analysis.

Author: A statement at the end of the Introduction was clarified to summarize the two-stage modeling approach (Line 140). The first stage estimates the plot's local environmental effect from the HTP, while the second stage leverages that estimate into genomic prediction either using random effects or fixed effects.

Comment 5. I had some trouble understanding what the authors mean by the term "local environmental effect (LEE)". This term sounds a bit cryptic. It would be much easier to understand if this term were introduced via a linear model being fitted. This could have an explicit effect for what the "LEE." My guess is that what the authors mean is simply the non-genetic effects associated with an observation / plot, i.e. possibly time-varying environmental effects for plots, blocks, replicates, as

opposed to the genotypic effects of interest. But a clarification, best via a statistical model and when the term is first introduced, would be very useful.

Author: We appreciate the concern about using the term "local environmental effect" (LEE) and so we opted to use the term "permanent environment" (PE) instead of LEE throughout the paper. We believe PE is a more standard term to describe these effects when modeling longitudinal data. We amended the explanation of Equation 1 to indicate the PE are solutions to the plot's specific effect (Line 299).

Comment 6. Equation 1: The residual "e" needs to be explained. What are the fixed effects in beta? What is a "local environment"? Isn't this just a plot, and is this why u_p carries the subscript "p"? Is this a model for a single time point or for multiple time points? Why does this model have no effects representing the randomization layout of the replicated field experiment (replicates, blocks)?

Author: We added the reference to the residual error in Model 1 and amended the explanation of Equation 1 to indicate the NDVI PE are solutions to the plot's specific non-genetic effect (Line 295-300). We added a clarification to indicate that the design replication nested with imaging event is represented in the fixed effects and how the different HTP time points are accounted (Line 295, 440, 460, 475).

Comment 7. In the text following Equation 1, and also that preceding Equation 3, the authors refer to the multivariate case in the case where multiple time points are considered for the same trait. I find the term "multivariate" misleading in this context, where the correlation that needs to be accounted for is one for serial or autocorrelation. Also, as the same trait is being measured at each time point, heterogeneity of variance is not necessarily an issue, as it would be with multiple traits. So perhaps the authors can reconsider their terminology here, also because their analysis ultimately focuses on the correlation between NDVI and agronomic traits of interest, which is what I would consider as a multivariate (multitrait) problem.

Author: We acknowledge the confusion that could come from the usage of the term multivariate when describing a model fitting many HTP time points, and we amended the manuscript to distinguish these models as either the "single-trait" case for fitting a single HTP time point or the "single-trait-repeated" case for fitting several HTP time points (Line 305, and throughout paper). The "multi-trait model" case of fitting HTP and agronomic traits in a single model is defined in Equation 6 (previously Equation 12). It should also be noted that, while these are repeated measurements, they are taken at different developmental stages of the plant and therefore would not be expected to have a uniform correlation structure or the same expression of genetic and residual variation. It would be expected that variance is heterogeneous across time and covariances would also change through time. To use an example from animal breeding, it is common to refer to birth weight, weaning weight, and weight at slaughter as separate but correlated traits.

Comment 8. Equation 4: It is not clear how this scalar equation ties in with the preceding models stated in matrix form. This is also because the notation looks completely different.

Author: We appreciate the concern in presenting different model notation, so we opted to only amend the text and use matrix notation throughout the main text. Some more detailed equations were moved to Supplemental File S2 and S3. With the references to published works in the text we hope to guide readers to further documentation.

Comment 9. Equation 5: The text purports that this equation give a structure for spatial correlation according to the 2DSpl model, but the structure clearly implies independence between all plots.

Author: We intended to show that the Sommer 2DSpl model created the design matrix Z_p in such a way as to model the row and column splines and the plot-level effects u_p are implemented as an identity covariance matrix. We moved this implementation detail to Supplemental File S2. A paper citation for the Sommer R package is included in the text. For more information please consult the spl2D documentation <https://www.rdocumentation.org/packages/sommer/versions/4.1.4/topics/spl2D>

Comment 10. Equation 6: The text here suggests that the model is for the "multivariate case", i.e. for the different time points, but the structure for Σ_{u_p} is diagonal, implying independence between time points, which is not a realistic model for repeated measurements.

Author: We appreciate comments in model selection. In the multi case of modeling several HTP at once, we leveraged genetic correlation across time points and looked at dynamic, independent plot-level effects for each HTP time point.

Comment 11. Equation 8: Same problem as with Equation 6.

Author: Similar with Equation 6 (now Equation S3), by letting the plot-level non-genetic effect for each HTP time point be independent we explored the dynamic nature of this effect across time. We used the random regression (RR) model as an alternative means of allowing for unstructured interactions across time points, and leveraged the RR model's ability to fit many time points at once.

Comment 12. Equation 9: This equation is for a further approach to spatio-temporal modeling. However, comparison with previously described models is made difficult by yet another change in notation. This equation is scalar. The model is announced to be for repeated measurements, but neither of the two random effects a_{kl} and p_{bl} is indexed by time. The only time-varying effects seems to be the residual $e_{ikn:t}$, so it is not clear how this model can account for serial correlation. Nor is it clear how this model accounts for spatial correlation. The description of the incidence matrices Z_a and Z_p would need to be given in detail for readers to appreciate how spatial and temporal correlation are accounted for. The authors just mention on page 11 that Legendre polynomials or linear spline functions can be used, but this is not enough. What exactly did the authors use as covariates? Certainly, there would need to be time indices involved in the scalar expressions in Equation 9 for this model to account for both spatial and temporal correlation. The matrix expressions given in the first paragraph on page 11 do involve subscripts for time, but the notation is not in agreement with Equation 9. So something is wrong here. The subscript for time itself has subscripts i and j , which is also confusing, because the subscript i has a different meaning in Equation 9. The fixed effect F_i is described as effect for the "replicate nested with image event data", but the effect is not indexed by t , which is at odds with this description. It is good, though, to have design effects in the model. Why is this not done for the other models as well? Later in the text, a matrix expression is given for the variance-covariance matrix of the response vector y . For this to be comprehensible, the ordering of scalar observations in the vector y would need to be given.

Author: We appreciate the concern and to simplify the text, we now only use matrix notation in the main text. Discussion of the random regression model has references to papers for further documentation. The text now shows more succinctly how spatial effects and RR permanent environment effects are related to the underlying plot-level permanent environment (PE) effect (Line 325-355).

Comment 13. The description of the structures for E on the lower part of page 11 is not succinct. Only one explicit equation is given for E_{ij} . For this second option, isotropy seems to be assumed. Is this a realistic assumption? What exactly are "positions"? The third to fifth option is cryptic. What are "empirical correlations"? How exactly are they computed? Explicit equations would help here. What is the underlying rationale?

Author: We altered the description to be more succinct and clarified that these are ordinal positions (Line 399). The empirical correlation matrices described in the third and fourth options were calculated using the plot-level soil measurements, while the empirical correlation matrices in the fifth and sixth options were calculated using the plot-level spatial effect estimates across the growing season. Therefore, plots which experienced similar spatial effects would empirically have higher correlations than those with dissimilar spatial effects. We added an equation in the text to explain the approach (Line 402). Isotropy is a common assumption for spatial models out of necessity. When there is only one measurement per plot this assumption allows partitioning of plot specific spatial effects from residuals. The use of longitudinal data enables this assumption to be relaxed, which is one of the

drivers for exploring these models. RRID would be an extreme case in which there are no assumptions about spatial relationships, albeit the spatial correlations are completely ignored in this case. The fact that it seems to perform well relative to models with strong assumptions of isotropy would suggest that it is indeed not a realistic assumption. The soil derived matrices remove this assumption while still accounting for spatial relationships, but make another strong assumption that specific soil properties are the main drivers of spatial patterns.

Comment 14. The description of the simulation on page 12 is very scant. It remains unclear from what model exactly the data are simulated. The same goes for the evaluation of accuracy, which is denoted as a five-step process. It mentions a first-stage model, but it is unclear what exactly a first-stage model is here. Also see my comment (4). In a second step, "the computed NLEE were subtracted from NDVI HTP in order to minimize latent spatio-temporal effects in the NDVI." This subtraction is never mentioned up to here, also not with the description of the various models. This would seem to be a key step in the two-stage approach announced in the introduction. Should this not be described as part of the modelling approach? Also, the subtraction will induce correlations among the corrected values. How are these taken into account in the downstream analysis? Ignoring them will lead to invalid inferences. For an illustration based on a simple case see Piepho, H.P., Williams, E.R., Ogotu, J.O. (2013): A two-stage approach to recovery of inter-block information and shrinkage of block effect estimates. *Communications in Biometry and Crop Science* 8, 10-22.

Author: We appreciate the opportunity to clarify this text. The simulation process was used to test the ability for the model to detect a plot's known local environmental effect, and therefore was not part of the agronomic genomic prediction piece. We have altered the text to remove the reference to the "first stage" and placed the statements more cohesively. Evaluating the simulation accuracy did not play a part in the larger part of this study, so the subtraction to minimize the latent effects did not occur outside of the simulation study. We amended the description on the simulation to be more concise (Line 420-430).

Comment 15. On page 13, the stage-wise approach is mentioned again. Specifically, a "first-stage NLEE" is mentioned. What exactly is this? It seems to be some kind of estimator derived from the models fitted to the NDVI data, but it seems this is not really explained anywhere. The subsequent Equation 13 used the empirical correlation among the NLEE over time to estimate the matrix L. An explicit equation for the correlation would be useful here. For the second option, based on "fixed factor effects (FE)", the key feature of the models seems to be a regression of the agronomic response on the NLEE as covariates. The description of the regression terms is a bit cryptic because it is not made explicit what forms the H and F matrices take and how NLEE features in these.

Author: We have clarified this in the text. The two-stage approach is summarized in the last paragraph of the introduction (Line 139). The first stage estimates the plot's local environmental effect (PE) using NDVI, while the second stage incorporates these estimates into genomic prediction as either random

effects or fixed effects. The matrix L was defined using the NDVI PE effects and we added the equation in the text. The text now uses bold to mean matrices versus vectors throughout.

Comment 16. Model performance is assessed via the heritability. The equation given seems to be on a plot basis. This should be made explicit and also justified. In field trials, heritability is usually assessed on a plot basis. Also, I am not sure the genetic variance can just be plugged in as done here because it is estimated from a model involving the GRM. How does this simple equation for heritability account for the pairwise genetic covariance among hybrids? And how does it account for the environmental covariance among residual errors "e"?

Author: We modified the text to indicate heritability was calculated on a plot basis (Line 545). The additive genetic variance is the variance component for the random effect of the lines and follows a genomic relationship matrix G . We amended the text to state these are variance components (Line 550).

Comment 17. Table 3: It is not clear how the correlations reported in this table were estimated. Is this based on a trivariate model for the three agronomic traits? It seems that the section "Agronomic Genomic Prediction" only describes univariate (single-trait) models. Even the "multi-trait model" in Equation 12 is only for one agronomic trait at a time.

Author: The correlations in Table 3 are between spatial effects estimated from the 2DSpline and AR1 models. Correlations for GY, GM, and EH were calculated individually and a trivariate model was not used. We modified the text to indicate these models were run independently for each trait (Line 582). The agronomic genomic prediction models, including the multi-trait model, used one agronomic trait at a time. We modified the text to distinguish between single-trait, single-trait-repeated, and multi-trait models more clearly (Line 304, and throughout text).

Reviewer 2:

In general, I liked the manuscript. It brings up an interesting topic and presents an array of modelling approaches to capture spatial heterogeneity in phenotypic variation in a secondary trait (vegetation index) and link that to spatial heterogeneity in target agronomic traits. I think the work can be very relevant as many research platforms are putting their trust in high-throughput phenotyping methods.

Having said that, I have some points of concern which I will try to explain below and which, as it happens, I believe become progressively more important.

Comment 1. One point that is missing is a discussion about the validity of the aerial images themselves as a proxy for spatial heterogeneity in local biomass/'green-up' effects. A UAV flying at a particular date may/does not differentiate true spatial heterogeneity from e.g. heterogeneity in phenological stages of different plants. Therefore what you see as spatial variation in the field may partly be variation in phenology rather than 'local spatial effects'. Not sure how relevant this problem is for prediction since you find local environment effects from the images (NLEE) to correlate with spatial heterogeneity in grain yield, but, judging from Fig. 2, correlations are not that high and NLEE appears to correlate weakly with soil parameters (Table 4). I therefore wonder to what extent we're looking at true spatial effects vs. different plant development stages. I guess it's an inherent caveat to HTP methodologies in general, but it would be worth discussing as this paper heavily revolves around spatial heterogeneity in the field.

Author: Thank you for the detailed discussion and review. We have amended the introduction to highlight the importance of using genomic relationships in our models for the HTP (Line 120). While physiological factors (e.g. greenness of different physiological stages) can be or are confounded in HTP like vegetation indices, the models presented in this study use genomic relationships to minimize such confounding between the additive genetic effects and spatial heterogeneity. Ultimately the performance of the models are evaluated by the accuracy in estimating agronomic trait effects, so any bias associated with confounding due to phenology that is specific to HTP would not improve accuracy for estimating genetic components of grain yield for example. Furthermore, the simulation piece in this study presented the ability of the tested models to recover induced spatial variability.

Comment 2. On the subject of spatial heterogeneity in the field, it is hard to judge from the text and figures how important spatial effects are in the studied trials. We are presented with spatial images of the field without the raw, phenotypic variation in the traits of interest as a reference. Also, if I'm correct the experimental design has never been mentioned; all I know is that there are two replicates (mentioned on p.15). The point is, in my experience working with spatial models in many a trial, most spatial variation is captured by the experimental design (e.g. by adding row/column effects to the model) and what spatial heterogeneity the spatial model picks up is usually quite small in relation to raw phenotypic variation. To put the importance of NLEEs into perspective I think the reader needs to be given (visual) guidelines as to the importance of these NLEEs in general.

Author: We have amended the text to include design information for the experiments (Line 153, 179). We appreciate the comment and have attempted to illustrate the importance of spatial heterogeneity in Figure 1 by including the magnitudes of the spatial effect in the heatmap (Line 572). We presented this approach to compare the two-dimensional spline against the simplest baseline model.

Comment 3. A more general point about the manuscript is that it's quite dense, with many models run and many results presented. I wonder if the manuscript could be tidied up a bit. For example, by adding a table that lists all models with their distinct random and fixed effects structures and a conceptual figure that shows how the first and second stage of the analyses are linked. Also, I wonder if the

number of models can be trimmed down a bit - or at least better justified/introduced. For example, random regression models (RRs) are invoked as an alternative to running spatial models at each time point. This seems like an appealing idea but could work only, in my view, if the curves can be fitted accurately. We don't get an impression reading the manuscript of how well the curves fit the data (3rd-order polynomials are used). If they don't fit well, we would a priori not expect the RRs to accurately predict spatial variation in the target trait, which is indeed what we see for grain yield in Figs. 4 and S8.

Author: We have tried to clarify the text by significantly changing the Methods section and moving some details to the Supplemental. Table 2 summarizes the models presented across the various fixed and random effects tested, and we amended the text to be clearer (Line 439, 460, 510). That equations now use matrix notation through the main text. The third order random regression model was selected by looking at the log-likelihood and estimated permanent environment effects with better correlations (as in Figure 4) than random regressions of first or second orders, and we amended this to reference studies into RR polynomials (Line 385).

Comment 4. A final point is that the authors should try to get the overall aim of the manuscript - or its value - as well as its conclusion more clear. The authors aim to investigate whether "local environment effects are able to detect spatial heterogeneity in agronomic traits" and whether these effects can aid in the genomic prediction. I'd say the latter is the most interesting part for breeders as we're ultimately interested if we can get accurate predictions, yet the manuscript puts in my view too much emphasis on 'getting the right spatial model'. We've already established that the correlation between local env. effects and spatial heterogeneity in target traits is not very strong; what really matters for breeders, I think, is how plot-level (or genotype-level, rather) values for intermediate phenotypes relate to values of the target trait. That comes in the second-stage analysis, but to my surprise there has been no cross validation. The authors basically looked at goodness of fit. For a true validation of your analyses you'd want to see how well your models do when predicting the phenotypes of 'unknown' genotypes. I think the authors need be clearer on the value of their analysis in this regard. This becomes also apparent in the conclusion section, which came across to me as somewhat inconclusive. This section mentions a lot of the model abbreviations and how they compared, but what is lacking a clear message, e.g. how we should go about leveraging HTP and spatial information to gain clearer predictions of our traits of interest.

Author: Thank you for the comment and we have amended the text to be more concise. Several studies have established the importance of spatial variability for agronomic traits, and this study observed significant variability in yield, grain moisture, and ear height that could be accounted for by in-field spatial heterogeneity. We agree that the manuscript has essentially two parts, first to explore the spatial heterogeneity in the HTP and agronomic traits using different models, and secondly to leverage the estimated spatial heterogeneity for spatial corrections. We have narrowed the text to focus on spatial corrections and not genomic prediction (Line 433). The metrics we chose to evaluate were heritability, model fit, and the correlation of genetic effects across replicates. We felt that looking at the correlation of genetic effects across replicates was most relevant because it assesses the extent to which we are able to account for spatial heterogeneity in the field. In future work we would like to assess cross

validation, but a dataset with a larger number of field experiments and a larger overlap in hybrids would be more suitable than what was used in this study.

Comment 5. My apologies for this long-winded review. I do like the ideas presented here and I like the variety of ways in which spatial effects are estimated; at this point, however, I think the manuscript is a bit too dense and lacking a clear goal. If the authors were somehow able to fix this it would make the manuscript much more interesting and relevant.

Author: We have made significant edits to the Introduction, Methods, and Results sections to hopefully clarify the models presented and the intended goals.

Comment 6. Some comments by section/page:

In general, please use subsections, especially in the methods section, to make it easier to navigate between the different analyses.

Author: We appreciate the comments presented. By rewriting the methods section and moving many extraneous details to the Supplemental we believe it is now easier to navigate the Methods section.

Comment 7. p.2, 2nd par: "...allow estimation of permanent environment effects from RR...". I think you need to introduce this concept a bit more. In animal breeding, a PE effect would indicate an effect of the animal on the phenotype independent of its genetics (e.g. food conditions during upbringing, body size, etc). Are you referring to the fact that individual plants (replicates) may consistently perform better or worse across time than other replicates from the same genotype? So if I'm correct, rather than estimating spatial effects across plots in the field at each time point you're estimating spatial heterogeneity as different longitudinal trajectories differing between individual plants/plots by fitting lines through time; you then look at the variation between longitudinal predictions at each time point (as per info on p11). I think a conceptual figure distinguishing the two approaches may be useful. But see my earlier comment on RR.

Author: Your understanding of how we intended to use the random regression (RR) permanent environment effects is correct. We fitted random regression lines for each plot through time, separated from the random additive genetic effects and therefore capturing non-genetic effects. We amended the text to be shorter and clarified that the RR effects are distinct from the spatial effects, but both are estimations of the specific plot's PE. We also now use the term "permanent environment" throughout the text to mean non-genetic, plot-specific environmental effects, estimated through spatial models or temporal RR models, and avoided adding the new "local environment effect" LEE term.

Comment 8. P4, Field Experiments: what is the design of the experiment? Later somewhere you mention there are two replicates.

Author: The field experiment design is now described including the design type and the number of genotypes tested in the different experiments (Line 153, 170).

Comment 9. P8, Eqn. 1: Please already give a brief explanation of the $Z_p u_p$ term. Also, how is the experimental design of the experiment factored into the model?

Author: We appreciate the comment and have clarified the text to describe how the experimental design is incorporated (Line 297).

Comment 10. P10, eqn. 9: See my comment above about the use of RR.

Author: The random regression model fits the permanent environment as lines for each specific plot across time. We amended the text to consistently use permanent environment (PE) to describe a plot's non-genetic local environment effect.

Comment 11. P11, mid: what does n stand for in the equation?

Author: The n stands for repeated observations on a given plot on a given time point; however, in this study only one observation per plot per time point was used. We now use matrix notation throughout (Line 439).

Comment 12. P11, same par: "...third order Legendre polynomials were considered". How well did these polynomials capture the spatial trend in HTPs? Would longitudinal spline curves not make more sense in these types of data?

Author: Third order polynomials were considered for the random regression after looking at the log likelihood of first and second order polynomials. We also looked at the correlation between the estimated HTP permanent environment effects and agronomic trait spatial heterogeneity. We edited the text to clarify this point (Line 385).

Comment 13. P13, eqns 11a-c: is there any other spatial correction to these models, based on experimental design?

Author: In those equations (now 5a-c), the fixed effects account for the replication in the design, while the row and column design information are used in the random spatial effects.

Comment 14. P14, table 2. Please make reference to the equations; it's at times hard to remember which model represents which.

Author: We appreciate concerns over the density of the number of models presented and have edited the Methods text to be more concise. Table 2 serves as a summary for all of the model abbreviations used and shows the pairings between first-stage local environmental effects with the second stage genomic prediction approaches. We believe there is value in presenting the eight distinct methods for estimating the spatial heterogeneity as well as the fixed and random effect implementations, particularly for parsimony with the soil models.

January 26, 2024

GENETICS-2023-306691

Spatio-temporal modeling of high-throughput multi-spectral aerial images improves agronomic trait genomic prediction in hybrid maize

Dear Dr. Morales:

Same two experts in the field have re-reviewed your revised manuscript, and I have read it as well. I am pleased to inform you that, with minor revisions, it is potentially suitable for publication in GENETICS. The reviewers have comments and concerns that need to be addressed in a revised manuscript. You can read their reviews at the end of this email.

It is important to consider all of their comments to improve overall presentation in your work.

We look forward to receiving your revised manuscript. Please let the editorial office know approximately how long you expect to need for revisions.

Upon resubmission, please include:

1. A clean version of your manuscript;
2. A marked version of your manuscript in which you highlight significant revisions carried out in response to the major points raised by the editor/reviewers (track changes is acceptable if preferred);
3. A detailed response to the editor's/reviewers' comments and to the concerns listed above. Please reference line numbers in this response to aid the editors.

Additionally, please ensure that your resubmission is formatted for GENETICS.

<https://academic.oup.com/genetics/pages/general-instructions>

Follow this link to submit the revised manuscript: Link Not Available

Sincerely,

Mikko J. Sillanpää
Associate Editor
GENETICS

Approved by:
Hongyu Zhao
Senior Editor
GENETICS

Reviewer #1 (Comments for the Authors (Required)):

The covariance structure for e in equation 1 needs to be specified right after the statement of the model.

The fixed effect β_{r_i} is repeatedly described as a block effect, nested within locations and years. This suggests that data of different locations and years were jointly analysed. In this case, the should the model not comprise GxL, GxY and GxLxY effects?

The authors are not specifying their heritability as on a plot basis. This may be okay for the equation given but I would question the relevance of this heritability. Selections are typically done on a genotype mean basis. So some justification is needed here. Also, the genetic variance can be taken at face value only if the GRM is properly scaled. Hence, the scaling of the GRM needs to be provided and justified.

Line 80: Only cite surnames of authors => delete "Arthur R."

Reviewer #2 (Comments for the Authors (Required)):

I have read the revised version of the manuscript and I think it has improved substantially, especially with regards to legibility. Some minor remarks:

In eqn 3, did you apply the unstructured VCOV matrix to all experiments, even 2020_NYH2 which has 12 timepoints? That seems unfeasible, unless we approximate it, e.g. using a factor-analytic structure. On a similar note, how were you able to run the RR models in experiments with only 4 timepoints? Perhaps I'm missing something but it would be good to spell this out. In general, there is still an abbreviation overkill which makes the manuscript hard to follow at times. For example, FE and L reappear in the text and tables/figures; it would be good to be occasionally reminded of their meaning (e.g. in figure/table captions. In general I would keep as a rule to spell terms in full if there is no need to abbreviate it.

Jip Ramakers

Associate Editor Comments:

Reviewer #1 (Comments for the Authors (Required)):

The covariance structure for e in equation 1 needs to be specified right after the statement of the model.

Done.

Lines 336 – 357 and Lines 409 -415

The fixed effect β_r is repeatedly described as a block effect, nested within locations and years. This suggests that data of different locations and years were jointly analysed. In this case, should the model not comprise $G \times L$, $G \times Y$ and $G \times L \times Y$ effects?

The models were run separately for each year as there is not much overlap in the lines between years. We have clarified the text.

Lines 300 - 311

The authors are not specifying their heritability as on a plot basis. This may be okay for the equation given but I would question the relevance of this heritability. Selections are typically done on a genotype mean basis. So some justification is needed here. Also, the genetic variance can be taken at face value only if the GRM is properly scaled. Hence, the scaling of the GRM needs to be provided and justified.

Justification for plot level heritability has been provided.

Lines 593 - 601

We have now provided details on the scaling of the GRM.

Lines 283-289

Line 80: Only cite surnames of authors => delete "Arthur R."

Corrected

Line 80

Reviewer #2 (Comments for the Authors (Required)):

I have read the revised version of the manuscript and I think it has improved substantially, especially with regards to legibility. Some minor remarks:

In eqn 3, did you apply the unstructured VCOV matrix to all experiments, even 2020_NYH2 which has 12 timepoints? That seems unfeasible, unless we approximate it, e.g. using a factor-analytic structure. On a similar note, how were you able to run the RR models in experiments with only 4 timepoints? Perhaps I'm missing something but it would be good to spell this out.

We have clarified that not all timepoints were used for the unstructured models. This has been clarified in the text.
Lines 354-357

Technically you can fit a polynomial regression on 4 times points. Whether or not it is robust is another question, but the results indicate that the functions were reliable, at least with the dataset at hand.

In general, there is still an abbreviation overkill which makes the manuscript hard to follow at times. For example, FE and L reappear in the text and tables/figures; it would be good to be occasionally reminded of their meaning (e.g. in figure/table captions. In general I would keep as a rule to spell terms in full if there is no need to abbreviate it.

Point well taken; however, at this point going back and removing abbreviations would likely result in introducing errors and inconsistencies. We have spelled out the abbreviations in the figure/table captions as suggested.
Lines 569-578; 647-648; 669-671; 706-708; 726-742; 766-770; 845-848; 877-886; 915-917

Associate Editor Comments:

February 18, 2024

RE: GENETICS-2024-306855

Dr. Kelly R. Robbins
Cornell University
Plant Breeding and Genetics
240 Emerson Hall
Ithaca, New York 14853

Dear Dr. Robbins:

Congratulations! We are delighted to inform you that your manuscript entitled "Spatio-temporal modeling of high-throughput multi-spectral aerial images improves agronomic trait genomic prediction in hybrid maize" is acceptable for publication in GENETICS. Many thanks for submitting your research to the journal.

To Proceed to Production:

1. Format your article according to GENETICS style, as discussed at <https://academic.oup.com/genetics/pages/general-instructions>, and upload your final files at <https://genetics.msubmit.net>.
2. Your manuscript will be published as-is (unedited-as submitted, reviewed, and accepted) at the GENETICS website as an Advanced Access article and deposited into PubMed shortly after receipt of source files and the completed license to publish. Please notify sourcefiles@thegsajournals.org if you do not wish to publish your article via Advanced Access.
3. We invite you to submit an original color figure related to your paper for consideration as cover art. Please email your submission to the editorial office or upload it with your final files. You can submit a small-sized image for evaluation, and if selected, the final image must be a TIFF file 2513px wide by 3263px high (8.375 by 10.875 inches; resolution of 600ppi). Please avoid graphs and small type.

If you have any questions or encounter any problems while uploading your accepted manuscript files, please email the editorial office at sourcefiles@thegsajournals.org.

Sincerely,

Mikko J. Sillanpää
Associate Editor
GENETICS

Approved by:
Hongyu Zhao
Senior Editor
GENETICS

note: Please add jnls.author.support@oup.com and genetics.oup@kwglobal.com (or the domains @oup.com and @kwglobal.com) to your email program's "safe senders" list. You will be contacted by both at various points during the production process.

Review comments (if applicable):